# Identification of the key role of IL-17RB in the treatment of osteoarthritis with Shaoyao Gancao decoction: Verification based on RNA-seq and bioinformatics analysis

Chengzhi Hou[1,2☯], Zhangjingze Yu[1,3☯], Qinghui Song[1☯], Xuelei Chu[3], Guangcheng Wei[3], Jia Zhu[1], Liping Yang[1], Yong Zhao[2], Ping Zhang[1]*, Qiuyue Li[1]*

1 Pharmacological Laboratory of Traditional Chinese Medicine, Wangjing Hospital, China Academy of Chinese Medical Sciences, Beijing, China, 2 Sports Medicine Department 2, Wangjing Hospital, China Academy of Chinese Medical Sciences, Beijing, China, 3 Department of Education, Wangjing Hospital, China Academy of Chinese Medical Sciences, Beijing, China

☯ These authors contributed equally to this work.
* liqiuyue1012@126.com (QL); pinglele@sina.com (PZ)

## Abstract

### Background

Shaoyao Gancao Decoction (SGD) is a classic and representative oral administration of traditional Chinese medicine formula. It is composed of two Chinese herbal medicines, *Paeoniae Radix Alba [Paeonia lactiflora Pall]* and *Glycyrrhizae Radix et Rhizoma*. The clinical study found SGD could effectively reduce clinical symptoms and improve the level of inflammation in osteoarthritis (OA) patients.

### Purpose

The aim of this study is to identify the efficacy and molecular mechanism of SGD in the treatment of OA, and find the new therapeutic target through RNA sequencing (RNA-Seq) to provide theoretical support for its clinical application.

### Methods

Destabilization of the medial meniscus (DMM) OA rat model was established in vivo. Hematoxylineosin staining, safranin O/fast green staining and immunohistochemistry were used to observe changes of cartilage Histology and extracellular matrix (ECM) of cartilage cells. In vitro, the chondrocyte-like cells were derived from ATDC5 cells and induced by interleukin-1 beta to establish the model. The medial meniscotibial ligament (MTT) test was used to identify the effects of SGD on chondrocyte-like cell proliferation, and immunocytochemistry was used to assess changes in chondrocyte ECM. The differentially expressed genes (DEGs) were obtained by RNA-Seq. Meanwhile, the core targets were found through bioinformatics analysis, and then verified by qRT-PCR and Western Blotting. The inflammatory factors IL-1β, IL-6 and TNF-α were detected by ELISA.

**Data Availability Statement:** All relevant data are within the paper and its Supporting Information files.

**Funding:** This research was supported by the Basic Research Project of Wangjing Hospital, China Academy of Chinese Medical Sciences (WJYY-YJKT-2022-04), Science and Technology Innovation Project of China Academy of Chinese Medical Sciences (CI2021A04901) and (CI2021A05054), the Fundamental Research Funds for the Central public welfare research institutes (ZZ13-YQ-036) and the National Natural Science Foundation of China (No.82174415) There was no additional external funding received for this study

**Competing interests:** NO authors have competing interests.

**Abbreviations:** SGD, Shaoyao Gancao Decoction; TCM, traditional Chinese medicine; OA, osteoarthritis; DMM, destabilization of the medial meniscus; ECM, extracellular matrix; DEGs, differentially expressed genes; RNA-seq, RNA sequencing; NSAIDs, nonsteroidal anti-inflammatory drugs; NGS, next generation sequencing; UHPLC-Q-TOF, ultrahigh-performance liquid chromatography coupled with quadrupole time-of-flight; SD, Sprague-Dawley; MMLT, medial meniscotibial ligament; H&E, hematoxylin and eosin; OARSI, Osteoarthritis Research Society International; COL2A1, collagen type II A1; MMP-13, matrix metalloproteinase-13; FBS, fetal bovine serum; ITS, insulin-transferrin-selenite; GAG, glycosaminoglycans; IL-1β, interleukin-1 beta; GO, Gene Ontology; KEGG, Kyoto Encyclopedia of Genes and Genomes; IL-17RB, interleukin 17 receptor B; IL-23R, interleukin 23 receptor; GDF5, growth differentiation factor 5; IL-6, interleukin 6; TNF-α, tumor necrosis factor alpha; PI3K, phosphoinositide 3-kinase.

## Results

SGD could alleviate cartilage degeneration, and reduce ECM degradation in OA by upregulating COL2A1 and downregulating MMP-13. 120 key targets were screened from DEGs by RNA-Seq. Based on further bioinformatics analysis, interleukin 17 receptor B (IL-17RB), interleukin 23 receptor and growth differentiation factor 5 were finally selected as core targets. IL-17RB has rarely been reported in previous studies about OA, and worthy of further study. Subsequently, it was found that the gene and protein expressions of IL-17RB were significantly reversed in model group after SGD treatment. Moreover, SGD could inhibit the release of inflammatory factors by mediating IL-17RB in OA.

## Conclusions

SGD reduced the release of inflammatory factors IL-1β, IL-6 and TNF-α, upregulated COL2A1 and downregulated MMP-13 to alleviate degradation of ECM, and reduced the cartilage degeneration and progression of OA by reducing IL-17RB in articular cartilage.

## Introduction

Osteoarthritis (OA) is the most common degenerative bone and joint disease in the world. According to epidemiological studies, the incidence rate of this disease increases significantly after age 60; the incidence among males is approximately 9.6%, and among females, it is approximately 18% [1,2]. OA has a serious impact on patients' quality of life due to a high disability rate. The main clinical symptoms include joint pain, stiffness, and limited mobility [3]. In recent years, with the aging of the global population and the rising rate of obesity, the incidence rate of OA has been increasing annually, which has imposed enormous economic burdens on society [4]. Osteoarthritis can be categorized into five stages.

At present, the treatment of OA primarily consists of nonpharmacologic and pharmacologic approaches. Among the pharmacologic treatments, options include oral nonsteroidal anti-inflammatory drugs (NSAIDs), topical NSAIDs, and other therapies, such as intra-articular treatment [5]. According to the guidelines, topical and oral nonsteroidal analgesics are considered first-line treatment options. As topical use of nonsteroidal drugs can result in higher local drug concentrations and lower systemic levels compared to oral administration, the reduction of symptoms is more pronounced with topical use than with oral administration [6].

However, NSAIDs are far from satisfactory as they can only temporarily relieve OA symptoms and have several side effects, such as increasing the risk of stomach and cardiovascular diseases [7]. Therefore, patients with a history of peptic ulcers should avoid using nonsteroidal anti-inflammatory drugs. If prolonged use is necessary, it is recommended to use antacids. Knee joint surgery is often used for patients in stages IV to V. However, surgery has disadvantages such as high cost, significant trauma, and numerous complications, and the lifespan of joint prostheses is limited, necessitating revision surgery. Therefore, it is urgent and necessary to develop a new, effective, and safe therapy for OA treatment.

For decades, oral administration of traditional Chinese medicine (TCM) formulas has been widely used in the treatment of OA in China, as it can restore joint function to a certain extent, enhance the overall effective rate, and reduce the recurrence rate of OA [8]. Shaoyao Gancao Decoction (SGD) is a classic and representative oral TCM formula recorded in the Treatise on Febrile Diseases, an ancient Chinese medicine book dating back to approximately 200 B.C.

SGD is composed of two Chinese herbal medicines, Radix Paeoniae Alba [Paeonia lactiflora Pall.] and Glycyrrhizae Radix et Rhizoma, in a 1:1 ratio [9]. In some clinical studies, SGD was found to effectively reduce clinical symptoms and improve the inflammation level in OA patients [10]. Previous pharmacological studies have shown that the two Chinese herbal medicines in SGD have a synergistic effect in improving joint pain and function in OA patients [11,12]. In addition, researchers have predicted the targets and pathways of SGD in OA treatment through network pharmacological methods and found that SGD may treat OA by regulating the cell cycle, apoptosis, drug metabolism, inflammation, and immune regulation [13]. However, due to the limitations of network pharmacology, the specific efficacy and pharmacological mechanism of SGD in OA treatment are still unclear, and its key targets and pathways need further identification and verification.

With the development of next-generation sequencing (NGS) technology, RNA sequencing (RNA-seq) has been widely utilized in various fields, such as human disease diagnosis, prognosis, and treatment [14,15]. Therapy with TCM formulas exhibits multicomponent, multitarget, and multiway characteristics. The popularization and application of RNA-seq have provided a new idea and method for elucidating its specific mechanisms [16,17]. In the present study, we aimed to identify the specific efficacy and molecular mechanism of SGD in the treatment of osteoarthritis (OA), as well as to search for new targets, in order to provide theoretical support for its clinical application. To achieve this goal, three sequential experiments were designed: 1) validation of the efficacy of SGD both in vivo and in vitro, 2) RNA-Seq and bioinformatics analysis for SGD, and 3) validation of the targets identified through RNA-seq. Ultimately, we selected IL-17RB as our target for research to explore whether SGD can mitigate osteoarthritis by alleviating articular cartilage degeneration through the regulation of IL-17RB.

## Materials and methods

### Reagents and materials

SGD is composed of *Radix Paeoniae Alba* (Beijing Sifang traditional Chinese medicine decoction pieces Co., Ltd, batch No.: 20111301, Origin: Anhui.) and *Glycyrrhizae Radix et Rhizoma* (Beijing Shengshilong Pharmaceutical Co., Ltd, batch No. 2012193, Origin: Inner Mongolia) at a ratio of 1:1. The crude drug of SGD was obtained from Wangjing Hospital pharmacy, China Academy of Chinese Medical Sciences (Beijing, PRC). Diacerein was purchased from Kunming Jida Pharmaceutical Co., Ltd. (Kunming, PRC).

Hematoxylin-eosin (H&E) was purchased from Baso (PRC). Safranin O/fast green was purchased from Sigma (USA). Fetal bovine serum (FBS) was purchased from Gibco (USA). Ethylenediaminetetraacetic acid (EDTA), toluidine blue, MTT solution and DMSO were purchased from Solarbio (PRC). Goat serum and anti-rabbit secondary antibody were purchased from Zsbio (PRC). Interleukin-1 beta (IL-1β) cytokine, ELISA kits, collagen type II A1 (COL2A1) and matrix metalloproteinase-13 (MMP-13) primary antibodies were purchased from Proteintech (USA). DMEM/F12 medium and penicillin-streptomycin were purchased from HyClone (USA). Insulin-transferase-selenite (ITS) was purchased from Cyagen (USA). TRIzol reagent was purchased from TIANGEN (PRC). Interleukin 17 receptor B (IL-17RB) primary antibody was purchased from Abcam (ab86488, 1:1000, USA).

### Preparation and quality control of Shaoyao Gancao Decoction

After soaking and decocting for 25 minutes, SGD was filtered to obtain the drug solution. The drug solution was concentrated using a rotary evaporator to a concentration of 2 g crude drug/ ml. SGD was stored in the refrigerator at 4°C after sterilization. In this study, the dose/concentration of SGD refers to the dose of crude drug. According to the equivalent dose conversion

between the human body and rats, 2.16 g/kg is the clinical equivalent dose and refers to the weight ratio between crude drug and rats. 1.08 g/kg, 4.32 g/kg and 8.64 g/kg are 1/2, 2 and 4 times the clinical equivalent dose, respectively.

In a previous study, our team analyzed the chemical composition of SGD through ultrahigh-performance liquid chromatography coupled with quadrupole time-of-flight (UHPLC-Q-TOF) [18]. Thirty-two constituents were tentatively characterized based on their retention times and MSn data, and four standard available constituents, including gallic acid, malic acid, hydroxy-ferulic acid, and isoliquiritigenin, were identified as quality control markers.

## Validation of the efficacy of SGD in vivo and in vitro

**Animals.**    In vivo experiments were conducted to test the effects of SGD in the treatment of OA. Female Sprague-Dawley (SD) rats (8 weeks old, 220±10 g), purchased from SPF Bio-technology Co., Ltd. (Beijing, PRC), were used for in vivo experiments. Animal certificate number: 110324201102334251. All animals were fed in the specific pathogen-free laboratory of Penney Testing Co., Ltd. (Beijing, PRC) and placed in plastic cages at a room temperature of 22±1˚C and a relative humidity of 50±10%.

**Establishment and treatment of the OA model in rats.**    Seventy female SD rats were randomly divided into 7 groups (10 rats per group): sham, model, positive drug and SGD (1.08, 2.16, 4.32 and 8.64 g/kg) experimental groups. After 1 week of adaptation, all rats were anesthetized by intraperitoneal injection of 3% pentobarbital. Subsequently, the rats (except the sham group) were subjected to operation-induced OA by destabilization of the medial meniscus (DMM)(S2 Table), including transection of the medial meniscotibial ligament (MMLT) and separation of the medial meniscus. The MMLT of sham group rats was only exposed but not transected. On the second day of the modeling operation, rats in the positive drug group were administered diacerein [19] (0.009 g/kg) and SGD (1.08, 2.16, 4.32 and 8.64 g/kg) in the experimental group and distilled water of equal amounts in the sham group and model group once a day for 8 weeks. All animals were euthanized after administration of SGD for 8 weeks. To alleviate suffering, rats were anesthetized intraperitoneally with 4% pentobarbital sodium (40mg/kg body weight) before modeling. At the end of the experiment, rats were again anesthetized intraperitoneally with 4% pentobarbital sodium. Under deep anesthesia, blood was collected from the abdominal aorta, leading to the animal's demise.

**Ethics statement.**    The animal study was performed following institutional guidelines for ethical animal studies and approved by the institutional Animal Ethics Committee, Penney Testing Co., Ltd. (PONY-2020-FL-65).

**Histology and immunohistochemistry.**    The samples from the tibial plateau of rats were fixed with 4% paraformaldehyde for 3 days at 4˚C and then decalcified with 10% EDTA (Solar-bio, PRC) at 4˚C for 3 weeks. Subsequently, the samples were embedded in paraffin and sectioned at a thickness of 5 μm. Finally, the sections were stained with H&E (Baso, PRC) and safranin O/fast green (Sigma, USA). The stained sections were photographed with a microscope (Olympus, Japan) and then scored using the Osteoarthritis Research Society International (OARSI) [20] assessment system in a blinded manner to evaluate the state of articular cartilage. For immunohistochemistry, sections were deparaffinized and rehydrated. The sections were blocked with 5% goat serum (Zsbio, PRC) and then incubated with anti-rabbit COL2A1 and MMP-13 primary antibodies (Proteintech, USA) overnight at 4˚C after antigen retrieval. Finally, the sections were incubated with anti-rabbit secondary antibody (Zsbio, PRC). Images were obtained by microscopy (Olympus, Japan) and analyzed by ImageJ software.

**Preparation of medicated serum.**    Twenty female SD rats were randomly divided into 2 groups (10 rats per group): the experimental group and the blank control group. The rats in

the experimental group were administered SGD (8.64 g/kg) once a day for 7 days, while the rats in the blank control group were administered an equal amount of distilled water. Blood was taken from the abdominal aorta 1 h after the final administration and then kept stationary at 4°C for 4 h. Blood serum was obtained by blood centrifugation at 3000 rpm for 10 min at 4°C. Subsequently, the blood serum was inactivated at 56°C for 30 minutes, filtered through a needle filter, and then stored at -80°C for subsequent experiments.

**Cell.** In vitro experiments were conducted to test the effects of SGD on OA. ATDC5 cells purchased from ATCC (USA) were used for in vitro experiments. ATDC5 cells were grown in normal culture medium containing DMEM/F12 medium (HyClone, USA), 10% FBS (Gibco, USA), and 100 U/ml penicillin-streptomycin (HyClone, USA). All cells were cultured as monolayers and maintained in a cell culture incubator at 37°C and 5% $CO_2$.

**Cell differentiation and treatment.** ATDC5 cells were induced by normal culture medium supplemented with 1% ITS (Cyagen, USA) for 21 days to obtain chondrocyte-like cells. The differentiation medium was replaced every two days. The glycosaminoglycan (GAG) content was stained with toluidine blue (Solarbio, PRC) and then photographed with a microscope (Olympus, Japan) on Days 0, 7, 14 and 21 to observe the GAG content. The differentiation degree was confirmed by the expression of COL2A1 and MMP-13 in immunocytochemistry. The optical density (OD) values of immunohistochemical images were calculated by ImageJ software. Chondrocyte-like ATDC5 cells were used for the following experiments. After culturing alone for 24 h, chondrocyte-like cells were pretreated with different concentrations of SGD-treated serum or diacerein for 1 h and then cotreated with the IL-1β (10 ng/ml, Proteintech, USA) cytokine for 24 h as the experimental group or positive control group. In addition, chondrocyte-like cells treated with IL-1β cytokine and 10% normal rat serum were regarded as the model control group, and only cells treated with 10% normal rat serum were regarded as the blank control group.

**Cell proliferation assay.** Chondrocyte-like cell proliferation was observed by the MTT test. In this trial, chondrocyte-like cells were transferred to 96-well plates at a density of $5 \times 10^3$ cells/ml. The control group included a blank control, model control and positive control (62.5, 125, 250 mg/ml diacerein). Treatment with SGD-treated serum (2%, 4% and 6%) was used as the experimental group. The culture medium was removed after treatment, and (3-(4,5-dimethylthiazol-2-yl)-2,5-diphenyltetrazolium bromide (MTT solution, Catalog #M1025, Solarbio, PRC) was added. The MTT solution was removed after 4 h, and 100 µl DMSO (Dimethyl sulfoxide, Solarbio, PRC) was added to each well. An ELISA meter was used to examine the absorbance of each well at a wavelength of 490 nm.

**Immunocytochemical.** The effect of SGD on extracellular matrix (ECM) degradation of IL-1β-induced chondrocyte-like cells was detected by immunocytochemistry. The cells were transferred to 24-well plates at a density of $1 \times 10^5$ cells/ml. The trial was divided into three groups: the blank control group, model control group and SGD (4% medicated serum) group. Subsequently, the culture medium was removed after treatment and fixed with paraformaldehyde for 20 min at 4°C. Next, the chondrocyte-like ATDC5 cells were incubated with $H_2O_2$ for 10 min at room temperature and then with anti-rabbit COL2A1 and MMP-13 primary antibodies (Proteintech, USA) overnight at 4°C. Finally, the method was the same as that used for previous immunohistochemistry.

## RNA-seq and bioinformatics analysis for SGD

Thirty female SD rats were randomly divided into 3 groups (10 rats per group): sham, model, and SGD (8.64 g/kg) experimental groups. As mentioned previously, rats in the model group and experimental group were induced by DMM, while those in the sham group were subjected

to sham operation. After the modeling operation, the rats in the experimental group were administered SGD (8.64 g/kg), while the rats in the sham group and model group were given equal amounts of distilled water once a day for 8 weeks. All animals were euthanized after administration of SGD for 8 weeks. The articular cartilage was removed from the distal femur and tibial plateau, frozen in liquid nitrogen immediately and stored at -80˚C for subsequent Experiments 2 and 3. Each sample used for RNA-Seq was extracted from 3 individuals to ensure an adequate RNA amount and to minimize individual differences. The RNA-seq and bioinformatics analyses were conducted by BioNovoGene Co., Ltd. (Suzhou, PRC).

Total RNA was extracted from cartilage, and the quality of the RNA was assessed by an Agilent 2100 bioanalyzer (Agilent Technologies, Germany) according to the manufacturer's instructions. cDNA libraries were constructed by reverse transcription of RNA fragments and then sequenced on an Illumina HiSeq 2500 (Illumina, USA) to obtain raw data. Subsequently, "Clean reads" were obtained by filtering out low-quality, adaptor-polluted and N (N indicates that the base information cannot be determined) reads in raw data. "Clean reads" were mapped to the reference genome using HISAT2 software (http://ccb.jhu.edu/software/hisat2/faq.shtml). Then, quantitation of gene expression was conducted. The differentially expressed genes (DEGs) were screened using DEseq2 [21] with |fold change| $\geq$ 1.00 and adjusted P value < 0.05 as the standard. The cluster analysis of DEGs among the groups is shown in the heat plot, while the overlap is shown in the Venn diagram. In addition, DEGs were also annotated to the Gene Ontology (GO) and Kyoto Encyclopedia of Genes and Genomes (KEGG) databases using clusterProfiler to explore the gene function and pathway, respectively.

## Validation of the targets identified by RNA-seq

**qRT-PCR.**   To validate the targets identified by RNA-seq at the mRNA level, the expression of IL-17RB, interleukin 23 receptor (IL-23R) and growth differentiation factor 5 (GDF5) in the sham, model, and SGD (8.64 g/kg) experimental groups was detected using qRT-PCR. The qRT-PCR method was the same as follows. In the qRT-PCR trial, total RNA from cartilage was extracted using TRIzol reagent (TIANGEN, PRC), the concentration and purity of which were detected using a Nanodrop One nucleic acid quantizer. Subsequently, the RNA was reverse transcribed into cDNA using PrimeScript RT Master Mix (Takara, Japan). As shown in Table 1, the primers were designed in the laboratory and synthesized by Sangon Biotech (Sangon, Shanghai, PRC) based on the mRNA sequences obtained from the NCBI database. Finally, qPCR was performed on an ABI7500 (Applied Bioscience, USA). mRNA expression is presented as the fold change relative to GAPDH and was determined using the $2^{-\Delta\Delta Ct}$ method.

**Western blotting.**   To validate the targets identified by RNA-seq at the protein level, the expression of IL-17RB in the sham, model, and SGD (8.64 g/kg) experimental groups was detected using Western blotting. The method of Western blotting was the same as follows. In the Western blotting trial, the protein was extracted from cartilage using lysis buffer, chilled and centrifuged at 12,000 rpm for 30 min at 4˚C. The protein concentration was estimated using the BCA method. The proteins were separated on a 10% SDS-PAGE gel and then

**Table 1.  The mRNA sequences of the 3 core targets obtained from the NCBI database.**

| Gene name | Forward primer | Reverse primer |
|---|---|---|
| Il17RB | AAGTGCTCCCTTCCCTCCAGATG | AGATGTCTTTGTGCTCCTTCCTTGC |
| Il23R | GGTAATATGTGGGTTGAGCCTGGTG | TGAAGATTCCTTGGTCGGCAGTTC |
| GDF5 | GCAGCATTACGCCATTCTTCCTTC | CCCTTTCTCCCGCACAACTGAC |
| GAPDH | CTCTGGTGGCTAGCTCAGAAA | CCCTGTTGCTGTAGCCGTAT |

transferred onto PVDF membranes. Subsequently, the membranes were blocked for 1 h at room temperature and probed with a primary antibody against IL-17RB (Abcam, ab86488, 1:1000, USA) overnight at 4˚C. The membranes were incubated with goat anti-rabbit horse-radish peroxidase-conjugated secondary antibody for 1 h at room temperature after washing with TBST. Finally, the membranes were observed by an enhanced chemiluminescence kit and quantified by ImageJ software. β-actin (Beyotime, AF2811, 1:1000, PRC) was used as an internal control.

**ELISA.** Inflammatory factors, including IL-1β, interleukin 6 (IL-6) and tumor necrosis factor alpha (TNF-α), were detected using ELISA in the sham, model, and SGD (8.64 g/kg) experimental groups. The articular cartilage samples of the right lower limb of rats were cut into pieces and ground with liquid nitrogen. Then, the lysate was added in proportion according to the tissue volume and broken with ultrasound. The supernatant was collected after centrifugation at 10000×g for 10 min at 4˚C. The bovine serum albumin standard was diluted in proportion, added to the 96-well plate with the sample protein and the working solution of butyleyanoacrylate, blended and incubated at 37˚C for 30 min. The expression levels of IL-1β, IL-6 and TNF-α were detected with ELISA kits (Proteintech, USA) according to the instructions of the manufacturer after detecting the absorbance of each well at a wavelength of 450 nm with a microplate reader.

## Statistical analysis

Data are presented as the mean ± standard deviation for at least three separate determinations for each group. Statistical analysis was performed using SPSS version 25.0 (IBM Corp.). Student's t test and one-way ANOVA were used to determine whether the difference among groups was statistically significant. #$p < 0.05$, ##$p < 0.01$, ###$p < 0.001$, *$p < 0.05$, **$p < 0.01$, ***$p < 0.001$ were considered to have statistical significance.(#represents the comparison of sham group with model group; *Represents the comparison of SGD group or dia group with model group.)

## Results

### The efficacy of SGD in vivo and in vitro

**SGD alleviates cartilage degeneration in OA.** In vivo, the rat knee cartilage from 7 groups, including the sham group, model group, diacerein group and SGD (1.08, 2.16, 4.32 and 8.64 g/kg) groups, was stained with H&E and safranin O/fast green to investigate the effects of SGD on the cartilage of DMM-treated rats. The results of H&E staining showed that compared with the sham group, the articular cartilage surface of rats in the model group was rough, and the chondrocytes were arranged irregularly. However, the articular cartilage surface of rats in the SGD (8.64 g/kg) and positive drug groups was smooth, and the chondrocytes were arranged in a line (Fig 1A). As shown in Fig 1B, safranin O/fast green staining demonstrated that the loss of proteoglycan was significant in the model group compared with the sham group, which was delayed by SGD (8.64 g/kg) and diacerein treatment. In addition, the staining results were supported by the OARSI score (Fig 1C). The above experimental results revealed that SGD could alleviate cartilage degeneration in OA.

**Differentiation of cells.** The ATDC5 cells induced by 1% ITS for 21 days were subjected to toluidine blue staining on Days 0, 7, 14 and 21 to observe the GAG content. As shown in Fig 2A, the staining intensity of the ATDC5 cells increased with the time of ITS induction, especially on Day 14. The toluidine blue staining results revealed that the GAG content of ATDC5 cells increased with ITS induction. Subsequently, the expression of chondrogenic differentiation markers (COL2A1 and MMP-13) was detected by immunocytochemistry on Days 0 and

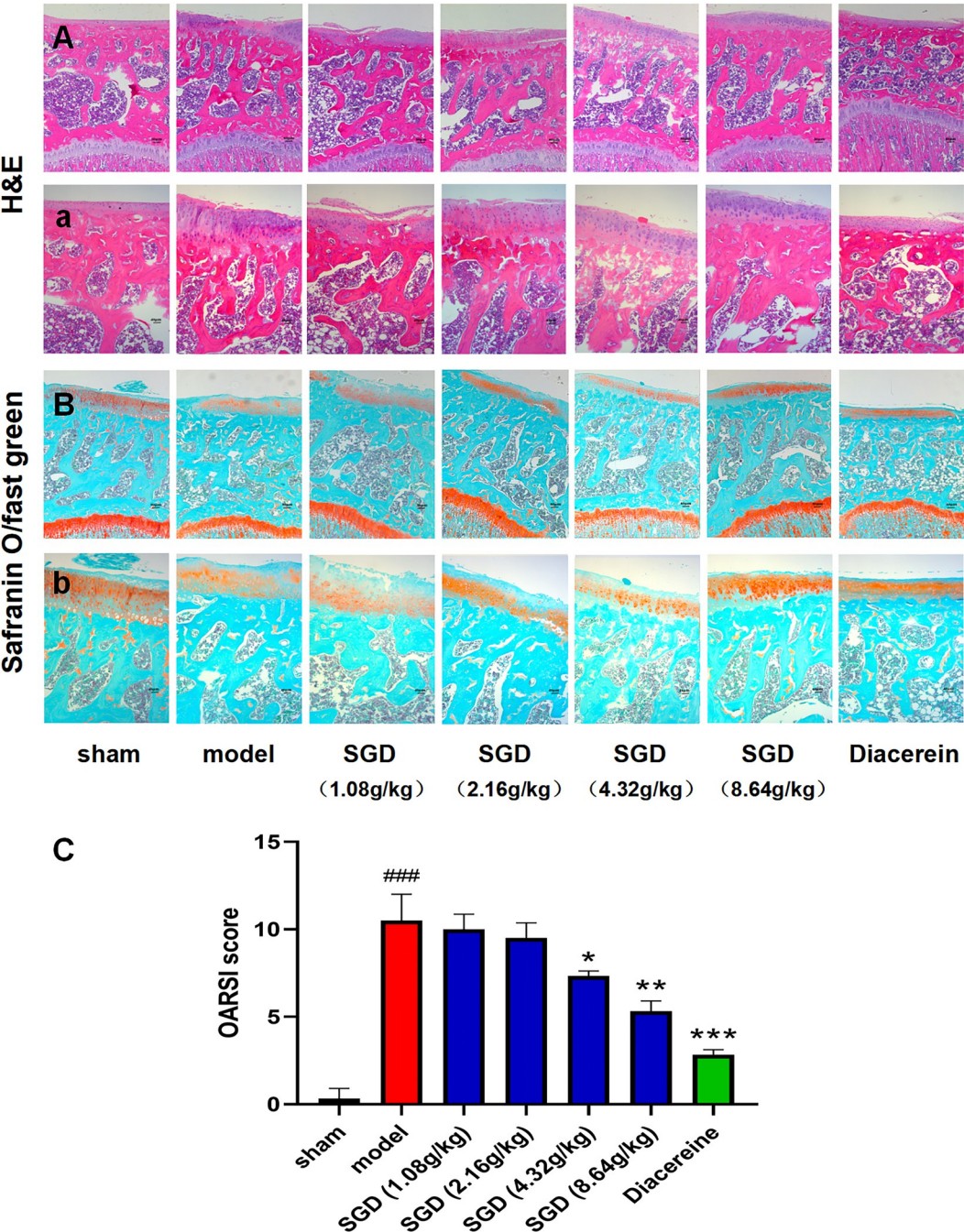

**Fig 1. SGD alleviated cartilage degeneration in DMM-induced OA.** After DMM modeling, rats were orally administered SGD at different concentrations (1.08, 2.16, 4.32 and 8.64 g/kg, weight ratio between crude drug and rats), diacerein (0.009 g/kg) or distilled water once a day for 8 weeks. **A.a.** H&E staining of the tibial plateau of rats (n = 3) Scale bar = 80 μm (A); Scale bar = 40 μm (a). **B.b.** Safranin O/fast green staining of the tibial plateau of rats (n = 3) Scale bar = 80 μm (B); Scale bar = 40 μm (b). **C.** OARSI scores. Data are expressed as the mean±SD, the sham group vs. the model group, #$p < 0.05$, ##$p < 0.01$, ###$p < 0.001$; the model group vs. the SGD or diacerein group *$p < 0.05$, **$p < 0.01$, ***$p < 0.001$. (one-way ANOVA were used).

14. As shown in Fig 2B, the staining intensity of ATDC5 cells (COL2A1 and MMP-13) on Day 14 was significantly higher than that on Day 0, and the OD values of the two groups were

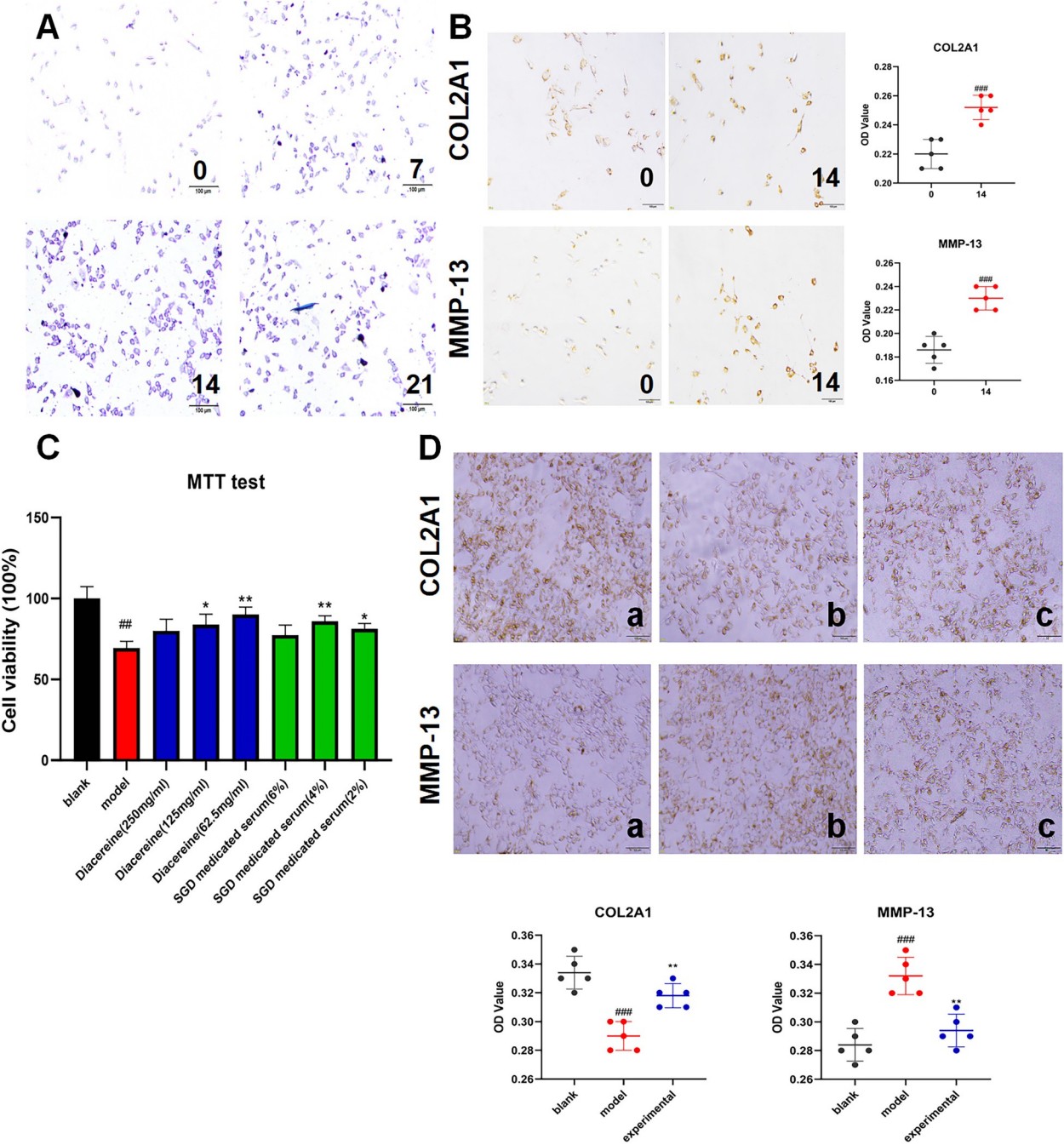

**Fig 2. The efficacy of SGD in vitro. A.** Toluidine blue staining of ITS-induced ATDC5 cells at different time points (0, 7, 14 and 21 days). **B.** Immunocytochemistry staining (COL2A1 and MMP-13) of ITS-induced ATDC5 cells on Days 0 and 1 (n = 5). **C.** Effects of SGD on chondrocyte-like cell proliferation (n = 6). **D.** Immunocytochemistry staining (COL2A1 and MMP-13) of IL-1β-induced chondrocyte-like cells after SGD-treated serum intervention: (a) blank group, (b) model group, and (c) experimental group (n = 5). Data are expressed as the mean±SD, the blank group vs. model or 14 day group, #$p < 0.05$, ##$p < 0.01$, ###$p < 0.001$, vs. the SGD, diacerein, experimental group *$p < 0.05$, **$p < 0.01$, ***$p < 0.001$. (one-way ANOVA were used).

significantly different ($P<0.05$). The immunocytochemistry results revealed that ITS induction could increase COL2A1 and MMP-13 content in ATDC5 cells. In conclusion, ATDC5 cells induced for 14 days differentiated into chondrocyte-like cells.

**Effects of SGD on chondrocyte-like cell proliferation.** In vitro, the effects of SGD on chondrocyte-like cell proliferation were assessed using the MTT test. There were four groups in these trials: the blank control group, model control group, positive control group (62.5, 125, 250 mg/ml diacerein) and experimental group (2%, 4% and 6% SGD medicated serum). As shown in Fig 2C, compared with the blank control group, the cell proliferation rate of the model control group significantly decreased ($P<0.05$). In addition, the proliferation rates of the positive control group (250 mg/ml diacerein) and experimental group (4% SGD-treated serum) were significantly higher than those of the model control group ($P<0.05$). The above findings suggest that SGD possesses the capability to counteract the suppressive action of the IL-1β cytokine on the proliferation of chondrocyte-like cells. Therefore, 4% SGD-treated serum was used as the best concentration for immunocytochemistry.

**SGD reduced ECM degradation in OA.** In vitro, the expression of COL2A1 and MMP-13 was detected by immunocytochemistry to observe the effect of SGD on ECM degradation of IL-1β-induced chondrocyte-like cells. The trial was divided into three groups: blank control group (a), model control group (b) and experimental group (c, 4% SGD medicated serum). As shown in Fig 2D, in the detection of COL2A1, the staining intensity and OD value of Group b were significantly lower than those of Group a, while those of Group c were significantly higher than those of Group b ($P<0.05$). In addition, in the detection of MMP-13, the staining intensity and OD value of Group b were significantly higher than those of Group a, while those of Group c were significantly lower than those of Group b ($P<0.05$).

In vivo, the effect of SGD on ECM degradation in DMM-induced rat cartilage was observed by immunohistochemistry (COL2A1 and MMP-13). The experiment was divided into 7 groups: sham group, model group, positive drug group (diacerein) and SGD (1.08, 2.16, 4.32 and 8.64 g/kg) groups. As shown in Fig 3A, the COL2A1 staining intensity in the model group was significantly lower than that in the sham group, and the COL2A1 staining intensity in the SGD (8.64 g/kg) and positive drug groups was significantly higher than that in the model group ($P<0.05$). As shown in Fig 3B, the staining intensity of MMP-13 in the model group was significantly higher than that in the sham group, and the staining intensity in the SGD (8.64 g/kg) and positive drug groups was significantly higher than that in the model group ($P<0.05$). In addition, the above staining results were supported by the OD value (Fig 3C and 3D).

In conclusion, SGD could reduce ECM degradation in OA by upregulating COL2A1 and downregulating MMP-13.

## RNA-seq and bioinformatics analysis data

As shown in Fig 4A, the volcano plots indicated the variation in mRNA expression among the groups (sham, model and SGD). According to |fold change| ≥ 1.00 and adjusted $P$ value < 0.05, 467 DEGs, including 285 upregulated and 182 downregulated DEGs, were screened in sham versus model cartilage (Fig 4B). In addition, Fig 4B also shows the number of DEGs in the model versus SGD comparison. The 120 DEGs overlapping between the two groups (sham vs. model and model vs. SGD) are shown in Venn plots and considered key targets (Fig 4C and S1 Table). The heat plot showed hierarchical clustering of DEGs among the groups (top 60, Fig 4D). Based on further bioinformatics analysis, Il-17RB, Il-23R and GDF5 from key targets were selected as core targets in the SGD treatment of OA, which were further verified by follow-up experiments.

All DEGs have been annotated to the GO and KEGG databases. A total of 868 GO items ($p < 0.05$) and 30 KEGG entries ($p < 0.05$) were collected. As shown in Fig 5A and 5B, the gene function of DEGs mainly included biological process, cellular component and molecular function, among which the most important were cartilage development and skeletal system

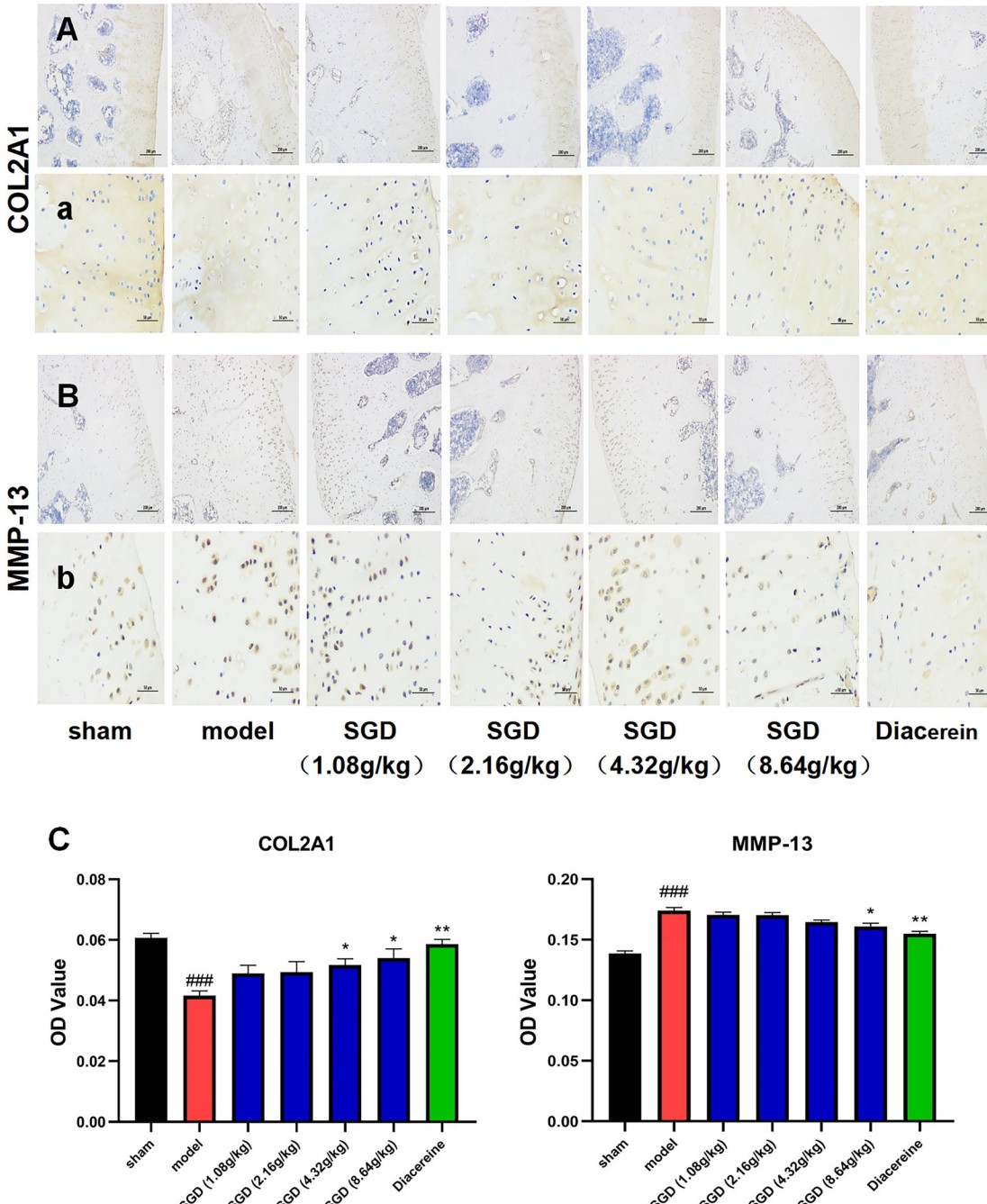

**Fig 3. SGD reduced ECM degradation in DMM-induced OA. A.** Immunohistochemistry staining (COL2A1) of the tibial plateau of rats (n = 3). **B.** Immunohistochemistry staining (MMP-13) of the tibial plateau of rats (n = 3). **C.** OD value of immunohistochemistry staining (COL2A1 and MMP-13). Data are expressed as the mean±SD vs. the model group, #p < 0.05, ##p < 0.01, ###p < 0.001 vs. the SGD or diacerein group *$p < 0.05$, **$p < 0.01$, ***$p < 0.001$. (one-way ANOVA were used).

development, extracellular region and extracellular matrix, extracellular matrix structural construct and signaling receptor binding, respectively. As shown in Fig 6A and 6B, the important pathways of DEGs were shown according to KEGG enrichment analysis, such as protein digestion and absorption, ECM-receptor interaction, cell cycle, glycosaminoglycan biosynthesis-heparan sulfate/heparin, phosphoinositide 3-kinase (PI3K)-Akt signaling pathway, cytokine-

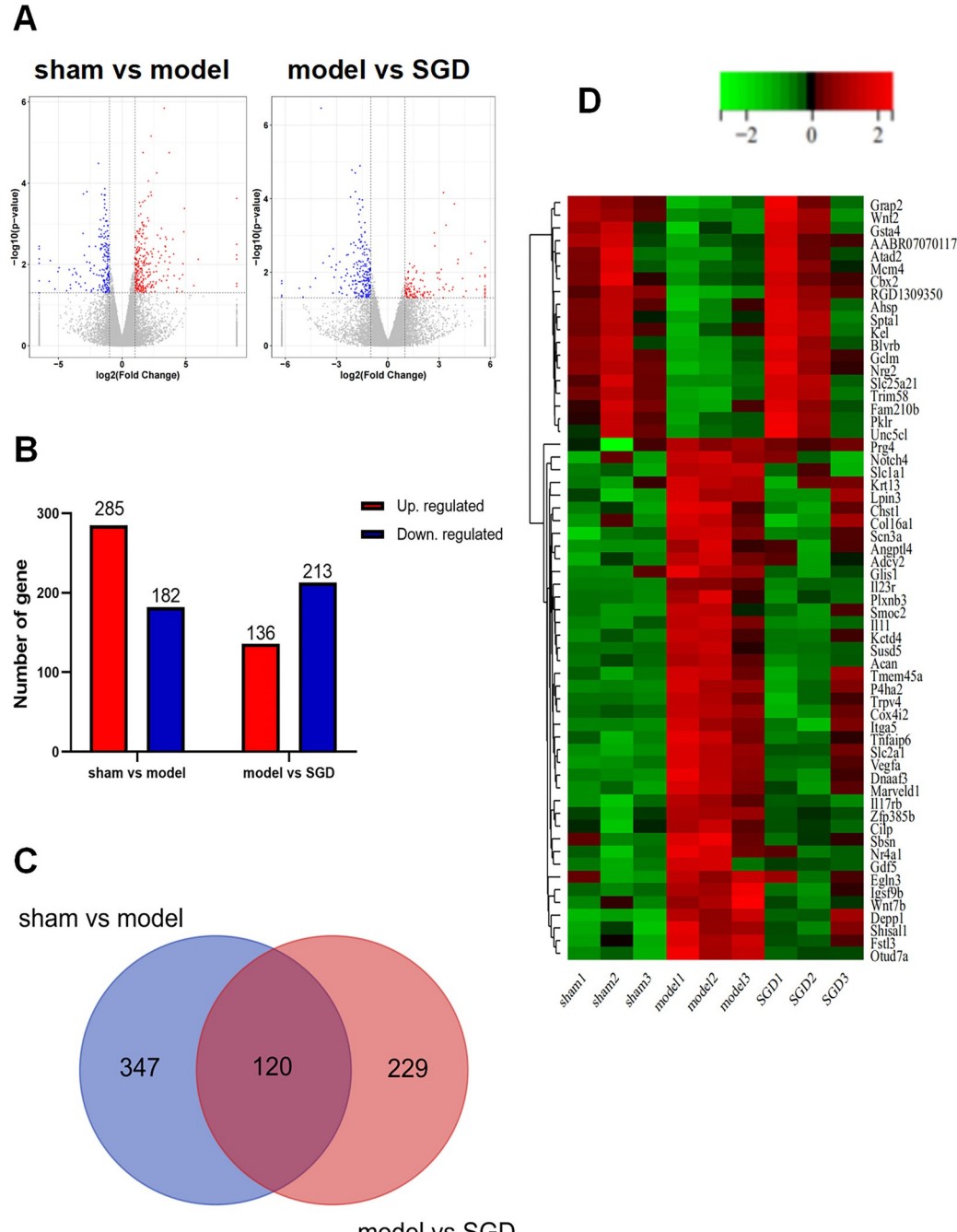

**Fig 4. RNA-seq and bioinformatics analysis data. A.** Differential expression gene volcano map. The red clusters represent upregulated genes, the blue clusters represent downregulated genes in the volcano plot, and the grey clusters represent no significant difference. **B.** The number of upregulated genes and downregulated genes between the normal group and the model group, as well as between the model group and the treatment group with SGD. **C.** Intersection genes between the normal group and the model group, as well as between the model group and the treatment group with SGD. **D.** Heat map analysis of intersecting genes (The columns represent samples, rows represent genes, red represents high expression, and green represents low expression). (Negative binomial distribution and one-way ANOVA were used).

cytokine receptor interaction and so on. Current studies have shown that the (PI3K)-Akt signaling pathway, extracellular matrix, IL-17, and others are closely related to OA [22–24]. The results of GO and KEGG enrichment analyses revealed highly OA-related gene functions and pathways.

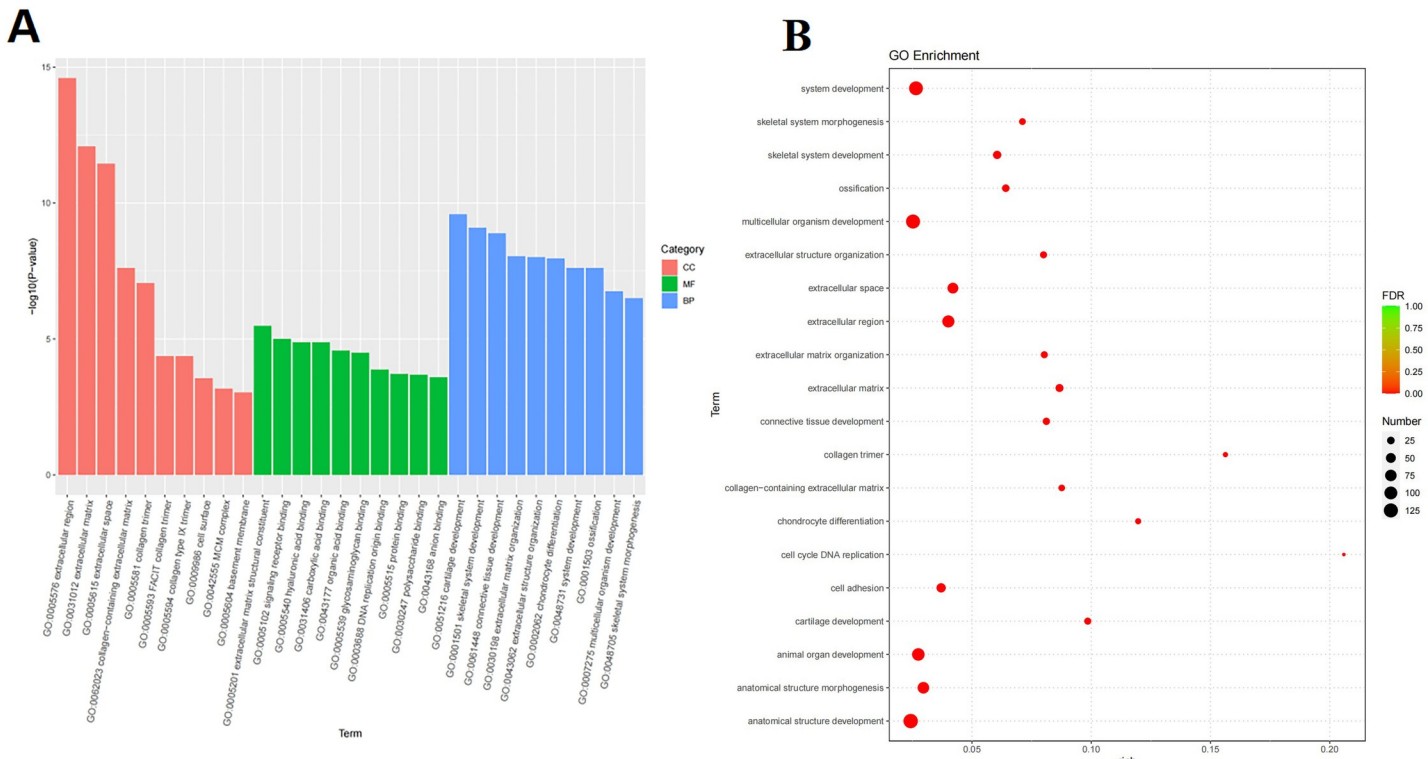

**Fig 5. Gene Ontology (GO) enrichment analysis of DEGs between the model group and SGD group. A.** Biological process (BP), cellular component (CC) and molecular function (MF) (top 10). **B.** Bubble diagram of GO analysis (top 20). (Negative binomial distribution were used).

### Validation of the targets identified by RNA-seq

**qRT-PCR and Western blotting analysis.** To verify whether the core targets identified by RNA-seq play important roles in SGD treatment of OA, the mRNA expression levels of Il-17RB, Il-23R and GDF5 were measured by qRT-PCR. As shown in Fig 7A, compared with that in the sham group, the mRNA expression of Il-17RB, Il-23R and GDF5 in the model group was significantly increased, and the trend was reversed after SGD treatment ($p<0.05$)(one-way ANOVA were used). Furthermore, the protein expression of Il-17RB by Western blotting analysis is shown in Fig 7B, and the change trend was consistent with the mRNA expression ($p < 0.05$)(one-way ANOVA were used).

In conclusion, the results showed that the above three core targets, especially IL-17RB, could play an important role in the treatment of SGD in OA.

**SGD reduced the release of inflammatory factors in OA.** To study the effect of SGD on inflammatory factors in OA, IL-1β, IL-6 and TNF-α in cartilage were detected by ELISA. As shown in Fig 7C, SGD treatment inhibited the increased IL-1β, IL-6 and TNF-α levels compared with the model group. The above results suggested that SGD could reduce the release of relevant inflammatory factors in OA.

## Discussion

OA, a public health problem of global import, is the leading cause of pain, loss of function and decreased quality of life in older individuals [25]. In recent years, people's understanding of OA has changed from simple joint wear to a more complex process composed of inflammatory and metabolic factors [26]. Inflammation, especially active synovitis, plays a key role in OA. It

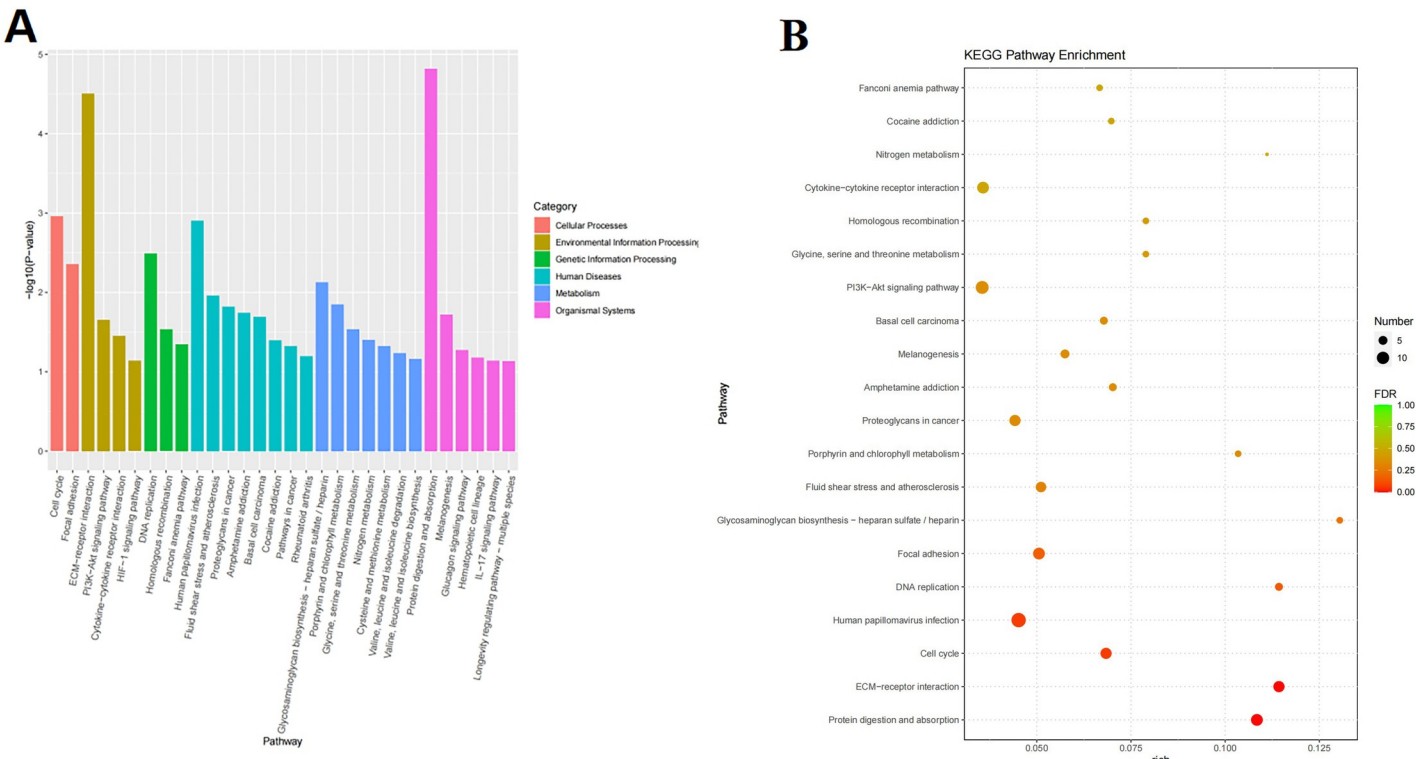

**Fig 6. KEGG enrichment analysis of DEGs between the model group and SGD group. A.** KEGG pathway (top 30). **B.** Bubble diagram of KEGG analysis (top 20). (Negative binomial distribution were used).

is widely accepted that the foreign body reaction caused by cartilage degradation in synovial cells leads to the production of metalloproteinases, synovial angiogenesis and inflammatory cytokines, resulting in the further degeneration of cartilage [27]. In addition, activated synovial macrophages also play an important role in OA [28]. Obviously, the association between OA and inflammation is a hot spot in the current research about the pathology of OA. In this study, the specific efficacy and molecular mechanism of SGD in the treatment of OA were identified. Meanwhile, core targets, including IL-17RB, IL-23R and GDF5, were found by RNA-Seq, in which IL-17RB was potentially associated with inflammation. Several studies have found that SGD has anti-inflammatory effects and inhibits the release of inflammatory factors in some diseases [29,30]. Therefore, the relationship between SGD and inflammation in the progression of OA is the focus of this study.

DMM-induced rats, a classical OA model, were selected for this in vivo experiment [31]. According to the results of HE staining, safranin O/fast green staining and the OARSI score in vivo, it was found that compared with the sham group, the rats after 8 weeks of modeling intervention had obvious articular cartilage degeneration, including different degrees of cartilage degeneration and chondrocyte necrosis, and indicated that the DMM-induced OA model was successfully established. The preservation of articular cartilage integrity in the group treated with 8.64 g/kg of SGD was markedly superior to that observed in groups receiving other dosages of SGD. It is well known that articular cartilage degeneration is the main pathological change in OA [32]. Therefore, it is considered that SGD could alleviate the degeneration of articular cartilage and delay the degeneration of articular cartilage in OA. ATDC5 cells, a mouse cartilage precursor cell, are widely used in the study of OA [33]. In vitro, chondrocyte-like cells are derived from ATDC5 cells and induced by IL-1β to establish a model [34]. The SGD-treated serum was

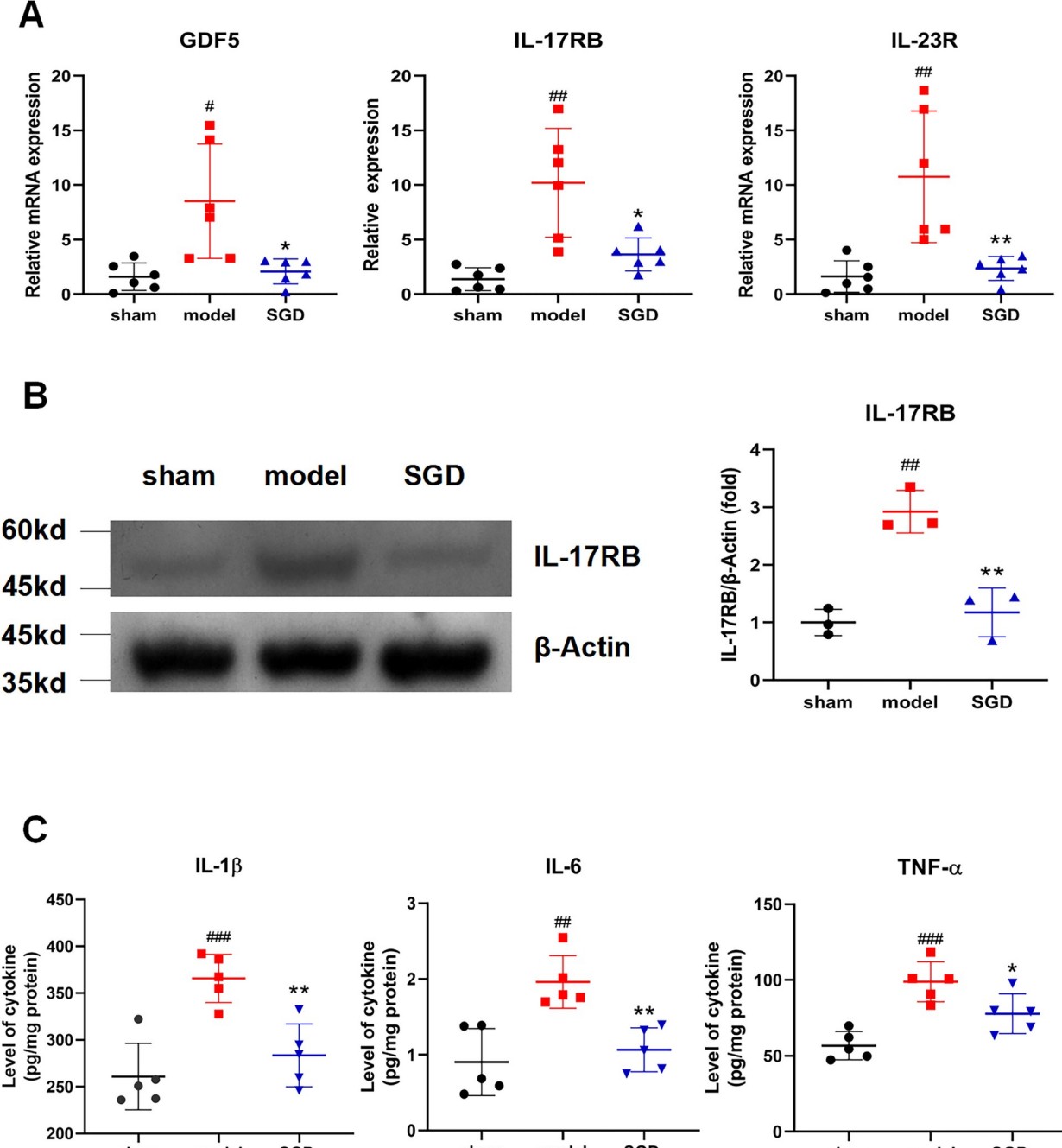

**Fig 7. Validation of the targets identified by RNA-seq. A.** qRT-PCR analysis (IL-17RB, IL-23R and GDF5) of cartilage from DMM-treated rats after SGD intervention (n = 6). **B.** Western blotting analysis (IL-17RB) (n = 3). **C.** ELISA analysis (IL-1β, IL-6 and TNF-α) (n = 5). Data are expressed as the mean±SD vs. the model group, #$p < 0.05$, ##$p < 0.01$, ###$p < 0.001$ vs. the SGD group *$p < 0.05$, **$p < 0.01$, ***$p < 0.001$. (one-way ANOVA were used).

prepared for the intervention of IL-1β-induced chondrocyte-like cells, and the MTT results showed that SGD could reduce the repression of IL-1β cytokines on chondrocyte-like cell proliferation. In addition, SGD-treated serum of 2%, 4% and 6% was screened with a cell viability greater than 50%, and 4% was the highest among the three concentrations in the MTT test. Therefore, 4% was the best concentration for the follow-up experiment.

It is known that the increase in matrix metalloproteinases (MMPs) in articular cartilage can disrupt the metabolic balance of ECM and accelerate the degradation of ECM [35]. As a representative collagenase, MMP-13 can degrade collagen type 2 (COL2) and proteoglycan, destroy the arrangement of collagen fibers, accelerate the cleavage of collagen components, and then lead to cartilage degeneration [36]. The immunohistochemical staining in vivo showed that the expression of MMP-13 in the articular cartilage of the model group increased significantly after modeling, while the expression of COL2A1 decreased, indicating that the content of MMP-13 in the articular cartilage of the rats increased after modeling and accelerated the degradation of COL2. The expression of MMP-13 in the SGD (8.64 g/kg) group was significantly lower than that in the model and other SGD groups, and the expression of COL2A1 showed the opposite trend. Therefore, it was found that SGD could reduce the degradation of ECM and degeneration of articular cartilage in OA. The in vitro immunocytochemistry results also supported this conclusion.

In this research, 120 targets were screened from DEGs by RNA-seq and bioinformatics analysis, in which Il-17RB, Il-23R and GDF5 were verified by qRT-PCR and Western blotting. The above three targets, especially IL-17RB, were potential core targets in the treatment of SGD in OA. Interleukin-17 (IL-17), which is mainly produced in Th17 cells, accelerates cartilage degradation by promoting MMPs and inflammatory cytokines such as IL-1β, IL-6 and TNF-α [37]. Meanwhile, IL-17A/IL-17RA can promote the degeneration of cartilage and small intestine via regulation of several inflammatory mediators in an OA murine model [38]. IL-17B, a new member of the IL-17 family, has 29% homology with IL-17A [39]. Several studies have found that IL-17B is a proinflammatory cytokine that is involved in the pathogenesis of inflammation [40]. IL-17RB, a member of the IL-17 receptor family, is activated by IL-17B and has been proven to be involved in inflammatory diseases [41,42]. Moreover, IL-17RB is found to exist in primitive and prehypertrophic chondrocytes in fibrous tissue [43]. In this experiment, it was found that SGD could reduce the release of relevant inflammatory factors, including IL-1β, IL-6 and TNF-α. Therefore, SGD could reduce the release of inflammatory factors by mediating IL-17RB to alleviate the progression of OA.

Several studies have shown that interleukin-23 (IL-23) plays a critical role in the production of IL-17 by activating Th17 cells [44]. Activation of the IL-23/IL-17 axis is an important target for inflammatory diseases [45,46]. Therefore, it is speculated that IL-23R may play an important role in SGD-mediated IL-17RB inhibition of inflammatory factors in the treatment of OA. IL-17E, also known as IL-25, is a critical regulator of type 2 immune responses and a driver of inflammatory diseases. It requires both IL-17 receptor A (IL-17RA) and IL-17RB to elicit functional responses. IL-25 primarily interacts with IL-17RB through two distinct linear and cyclic epitopes, leading to the activation of the nuclear factor kappa-light-chain enhancer of activated B cells (NF-κB). This interaction is crucial for its pro-inflammatory effects, which play a significant role in the pathogenesis of Th17-predominant diseases[47–49].

GDF5, a major risk locus for OA, has been widely studied in the treatment of OA in recent years [50]. Several studies have shown that the expression of GDF5 in articular cartilage is upregulated after DMM modeling, which is consistent with our experimental results [51]. The above results suggest that IL-23R and GDF5 may be potential targets for SGD in the treatment of OA, which is worth further study in the future.

## Conclusion

In the present study, the specific efficacy and molecular mechanism of SGD in the treatment of OA were revealed by three experiments. In addition, IL-17RB, IL-23R and GDF5 targets correlated with OA treatment, of which IL-17RB was the most critical. In contrast to GDF5,

IL-17RB has rarely been reported in previous studies on OA treatment, which is also the largest innovation of this experiment. In summary, SGD can reduce the release of inflammatory factors and the degradation of ECM by mediating IL-17RB to alleviate cartilage degeneration and progression of OA in vivo and in vitro. This conclusion also provides theoretical support for the clinical application of SGD in the treatment of OA.

## Supporting information

**S1 Table. Supplementary table.**
(XLSX)

**S2 Table. The specific modeling process of DMM.**
(DOCX)

**S1 Graphical abstract.**
(TIF)

## Acknowledgments

We'd like to acknowledge Suzhou Panomix (https://www.panomix.com/) and Jinyu Gu for providing help.

## Author Contributions

**Conceptualization:** Yong Zhao, Ping Zhang, Qiuyue Li.

**Data curation:** Chengzhi Hou, Zhangjingze Yu, Qinghui Song, Xuelei Chu, Jia Zhu, Liping Yang, Qiuyue Li.

**Formal analysis:** Chengzhi Hou, Zhangjingze Yu, Qinghui Song, Xuelei Chu, Guangcheng Wei, Ping Zhang, Qiuyue Li.

**Funding acquisition:** Yong Zhao, Ping Zhang, Qiuyue Li.

**Methodology:** Chengzhi Hou, Qinghui Song, Yong Zhao, Ping Zhang, Qiuyue Li.

**Software:** Chengzhi Hou, Qinghui Song.

**Supervision:** Yong Zhao, Qiuyue Li.

**Writing – original draft:** Chengzhi Hou, Zhangjingze Yu, Qinghui Song, Xuelei Chu.

**Writing – review & editing:** Chengzhi Hou, Zhangjingze Yu, Xuelei Chu.

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
