## [Decision Letter · Decision Letter 0]

2 Oct 2024

PONE-D-24-35164Identification of the key role of IL-17RB in the treatment of osteoarthritis with Shaoyao Gancao decoction: Verification based on RNA-seq and bioinformatics analysisPLOS ONE

Dear Dr. Yu,

Thank you for submitting your manuscript to PLOS ONE. After careful consideration, we feel that it has merit but does not fully meet PLOS ONE’s publication criteria as it currently stands. Therefore, we invite you to submit a revised version of the manuscript that addresses the points raised during the review process.

We look forward to receiving your revised manuscript.

Kind regards,

Xindie Zhou

Academic Editor

PLOS ONE

Journal Requirements:

2. To comply with PLOS ONE submissions requirements, in your Methods section, please provide additional information regarding the experiments involving animals and ensure you have included details on (1) methods of sacrifice, and (2) efforts to alleviate suffering.

“This research was supported by the Basic Research Project of Wangjing Hospital, China Academy of Chinese Medical Sciences (WJYY-YJKT-2022-04) , Science and Technology Innovation Project of China Academy of Chinese Medical Sciences (CI2021A04901) and (CI2021A05054), the Fundamental Research Funds for the Central public welfare research institutes (ZZ13-YQ-036) and the National Natural Science Foundation of China (No.82174415)”

“This research was supported by the Basic Research Project of Wangjing Hospital, China Academy of Chinese Medical Sciences (WJYY-YJKT-2022-04) , Science and Technology Innovation Project of China Academy of Chinese Medical Sciences (CI2021A04901) and (CI2021A05054), the Fundamental Research Funds for the Central public welfare research institutes (ZZ13-YQ-036) and the National Natural Science Foundation of China (No.82174415).**”**

“This research was supported by the Basic Research Project of Wangjing Hospital, China Academy of Chinese Medical Sciences (WJYY-YJKT-2022-04) , Science and Technology Innovation Project of China Academy of Chinese Medical Sciences (CI2021A04901) and (CI2021A05054), the Fundamental Research Funds for the Central public welfare research institutes (ZZ13-YQ-036) and the National Natural Science Foundation of China (No.82174415)”

5. We note that your Data Availability Statement is currently as follows: [

All relevant data are within the manuscript and its Supporting Information files.]

7. Please amend your authorship list in your manuscript file to include author Di Xia.

8. Your ethics statement should only appear in the Methods section of your manuscript. If your ethics statement is written in any section besides the Methods, please move it to the Methods section and delete it from any other section. Please ensure that your ethics statement is included in your manuscript, as the ethics statement entered into the online submission form will not be published alongside your manuscript.

8. Please upload a copy of Supporting Information Table1 which you refer to in your text on page 23.

Reviewers' comments:

Reviewer's Responses to Questions

**Comments to the Author**

1. Is the manuscript technically sound, and do the data support the conclusions?

Reviewer #1: Partly

Reviewer #2: Yes

Reviewer #3: Partly

2. Has the statistical analysis been performed appropriately and rigorously? 

Reviewer #1: Yes

Reviewer #2: Yes

Reviewer #3: No

3. Have the authors made all data underlying the findings in their manuscript fully available?

Reviewer #1: No

Reviewer #2: No

Reviewer #3: No

4. Is the manuscript presented in an intelligible fashion and written in standard English?

Reviewer #1: Yes

Reviewer #2: Yes

Reviewer #3: Yes

5. Review Comments to the Author

Reviewer #1: This is a well-written article. However, the experiments conducted do not fully support the conclusions presented in this paper. The discussion section is rather brief, which is unfortunate, as you could have explored the IL-17B signaling cascades based on bioinformatics analyses. I found the bioinformatics analyses to be quite standard, and I would have appreciated seeing co-expression profiles of IL-17RB, IL-23R, and GDF5. I was somewhat surprised not to see key genes involved in the IL-17B signaling pathway differentially expressed in Figure 4-C. Additionally, the biological replicate profiles of SGD3 show considerable variation, which raises concerns about the statistical significance of the expression analysis. Finally, to confirm the key role of IL-17RB, an experiment specifically expressing IL-17RB in sham, model, and SGD conditions would be necessary.

Reviewer #2: Major Comments:

1. Lack of Clinical Application Outlook:

The study’s results are derived from animal and cell models, making it unclear how the findings could translate into human clinical practice. The paper does not sufficiently discuss the feasibility of applying Shaoyao Gancao Decoction in human trials, nor does it explore potential side effects or safety concerns.

2. Weak Interpretation of Data and Statistical Significance:

The explanation of statistical methods and significance is weak in some sections. The lack of clear statistical methodology makes it difficult to fully assess the reliability of the conclusions.

Minor Comments:

1. Clarification of Expression:

Page 2, Lines 10-12: The phrase "Shaoyao Gancao Decoction reduced cartilage destruction and suppressed inflammatory factor release" is vague. A clearer explanation of the mechanism is needed.

2. Figure Readability:

Page 10, Lines 3-7, Figure 2: The color contrast in the figure is weak, making it hard to distinguish between the different groups.

3. Inconsistent Terminology:

Page 5, Lines 15-17: The term "cartilage destruction" is used inconsistently with terms like "cartilage degeneration" and "cartilage damage." Use consistent terminology throughout.

4. Grammar Corrections:

Page 3, Lines 23-24: The sentence is missing a verb. The sentence structure needs to be revised for clarity.

5. Reference Formatting:

Page 14, Lines 10-12: Some references are incomplete or incorrectly formatted.

Reviewer #3: ABSTRACT

The phrase “……..induced by interleukin-1 beta to establish model” in the abstract should be “………….induced by interleukin-1 beta to establish the model”

The phrase “…………immunocytochemistry to change of chondrocyte ECM” in the abstract is not clear. Do the authors mean changes in chondrocyte ECM or what?

The phrase “……Based on literature research” in the abstract cannot come up in the result section. Most likely in the discussion section

Some terminologies like OA and MTT test were first not introduced before use of the abbreviations

INTRODUCTION

Authors did not specify the grade of OA that is amenable to medical treatment and also left out surgical care. Also NSAIDs is not the only drugs available for treating OA and in practise NSAIDS is avoided in patients with history of Peptic ulcer disease and antacids are recommended if NSAIDs is going to be for prolong use. Hence complications of NSAIDs as the justification for the study is not strong enough as this can be avoided. And even in the reference that was cited (ref 5), only topical NSAIDs was recommended.

The aim from the title is to identify the key role of IL-17BR in the treatment of OA and there was nowhere this was mentioned in the introduction. Authors should please rectify

METHODOLOGY

The reason for use of diacerein is not explained in the methodology

What is MTT solution and what is it used for? What is DMSO? not explained before use of abbreviation

Diacerein was mentioned without stating the reason for its use in this study

Diacerein was not stated to be used as a positive drug. It makes the methodology very confusing

The statement “……All animals were euthanized after administration of SGD for 8 weeks” is confusing. Is it all animals or those that were administered SDG alone?

In section 2.3.3, what does “….in a blinded manner “ mean? This is a very ambiguous statement and needs further clarification. Also, the statement “…then scored using the Osteoarthritis Research Society International (OARSI) assessment” should be referenced

In section 2.3.4, are the twenty rats part of the initial 70 rats or new set of rats? This should be clarified.

In section 2.3.7, the statement “…The culture medium was removed after treatment, and MTT solution (Solarbio, PRC) was added. The MTT solution was removed after 4 h” is confusing. Please clarify. Also what is the use of the MTT solution? Please explain.

In section 2.3.8, while was only the 4% SDG used at this stage? Also, the statement “….Finally, the method was the same as that used for

previous immunohistochemistry” is not clear. Please expalin

In section 2.3.6, what is OD values?

In section 2.5.1, the cytokines identified in the RNA-seq were mainly focused on probably as a result of literature search, but this was never brought up in the introduction. Or why was the focus mainly on the expression of IL-17RB, interleukin 23 receptor (IL-23R) and growth differentiation factor 5 (GDF5)

The footnote to fig 1 is not clear: there is A,a,B,b. what is the picture in a and b showing? It would have been nice to annotate what the authors want us to see because of readers who are not histologically inclined.

In the footnote to fig 1, the statement “….Data are expressed as the mean±SD vs. the model group, #p < 0.05, ##p < 0.01,

###p < 0.001 vs. the SGD or diacerein group *p < 0.05, **p < 0.01, ***p < 0.001.” is not clear. The type of plot should be stated (barplot with error bar) Though not stated, this looks like a one-way ANOVA post-hoc analysis. The result of the statistical significance was as a result of which group comparison? Or if the analysis is based on a planned comparison, the authors should please state

Without knowing which group comparison yielded the result presented in fig 1, the statement “……The above experimental results revealed that SGD could alleviate cartilage

destruction in OA” is controversial. For example if the statistical significance seen in the SDG(8.64kg/l) is as a result of comparison with the sham group, that is explicit, but if the result is as a result of comparison with diacerein, then the interpretation is controversial because both seem to preserve the synovial joint

The presentation of the result can be improved on.

Fig 2c has same issue as fig 1. In fig 2c the height of the barplot are obviously not too different from each other which means whatever significance was detected was just statistical but not clinical. The actual data in figures should be presented

The conclusion of fig 3 is controversial: both the model group, positive drug group and experimental group showed downregulation and upregulation. Could this be a case of statistical significance and not clinical significance. The actual data (OD value) should be presented in a table for clarity

In section 3.1.2, 0, 7, 14, and 21 days were mentioned which were not in section 2.3.6. was this an afterthought? Mentioning of the days should come under methodology and not in result section

In subsection 3.1.3, the statement “……..As shown in Fig. 2C, compared with the blank control group, the cell proliferation rate of the model control group significantly decreased (P<0.05). In addition, the proliferation rates of the positive control group (250 mg/ml diacerein) and experimental group (4% SGDtreated serum) were significantly higher than those of the model control group (P<0.05)” is not a scientific way of presenting the result. What is the values compared and what is the confidence interval. This should be stated along with the P-value and not just the P-value alone

Subsection 3.1.4, same issue as subsection 3.1.3. the result should of the P-value should be presented more scientifically. As presently presented, there may just be statistical significance but no clinical significance

Under subsection 3.2, the statement “…….Based on literature research and further bioinformatics

Analysis” is very ambiguous and usually should not be in the result section. What is the result of the analysis under this subsection?

One cannot make out the labelling of fig 5 and 6 and hence makes it difficult to interpret. Fig 5 and fig 6 should be made legible to facilitate easy interpretation.

In section 3.3.2, the statement “….SGD treatment inhibited the increased IL-1β, IL-6 and TNF-α levels in the model group” is confusing. Please clarify

Fig 4 and its legend is not clear making interpretation difficult

It is known that data from DNA/RNA sequencing and cytokines can be skewed and hence data transformation is normally recommended before use of such data especially for parametric test were normality of the data is a perequisite. Were the RNA sequence data transformed before analysis. The authors should show that the continuous data analysed in this data have normal distribution or were transformed appropriately to achieve normality to justify use of the T-test.

All the places where P-value was mentioned should mention the test used, the values compared and the confidence interval and not just the P-value alone.

DISCUSSION

The statement “……the integrity of articular cartilage in the SGD (8.64 g/kg) group was significantly better than that in the other groups” is misleading and should be rephrased. Fig 1 actually showed that diacerein showed more significant results compared to others

The statement “…The expression of MMP-13 in the SGD (8.64 g/kg) group was significantly lower than that in the model and other SGD groups, and the expression of COL2A1 showed the opposite trend. Therefore, it was found that SGD could reduce the degradation of ECM and degeneration of articular cartilage in OA” is controversial. This could just be a case of statistical significance and not clinical significance because similar result was shown in the model group.

6. PLOS authors have the option to publish the peer review history of their article (what does this mean?). If published, this will include your full peer review and any attached files.

Reviewer #1: No

Reviewer #2: No

Reviewer #3: **Yes: **Dr Oluwasegun Adedeji Aremu

---

## [Author Response · Author response to Decision Letter 0]

6 Nov 2024

Dear Editor,

Thank you for your insightful feedback and constructive suggestions. We have meticulously revised the manuscript with the changes highlighted in blue for easy reference. Below, we address your comments.

Thank you for your reminder. We have completed the modifications according to the format requirements.

2.To comply with PLOS ONE submissions requirements, in your Methods section, please provide additional information regarding the experiments involving animals and ensure you have included details on (1) methods of sacrifice, and (2) efforts to alleviate suffering.

Thank you for your suggestion. We have truthfully added specific methods for animal euthanasia and how to alleviate suffering in the corresponding section of the article, as follows:

To alleviate suffering, rats were anesthetized intraperitoneally with 4% pentobarbital sodium (40mg/kg body weight) before modeling. 

At the end of the experiment, rats were again anesthetized intraperitoneally with 4% pentobarbital sodium. Under deep anesthesia, blood was collected from the abdominal aorta, leading to the animal's demise. 

“This research was supported by the Basic Research Project of Wangjing Hospital, China Academy of Chinese Medical Sciences (WJYY-YJKT-2022-04) , Science and Technology Innovation Project of China Academy of Chinese Medical Sciences (CI2021A04901) and (CI2021A05054), the Fundamental Research Funds for the Central public welfare research institutes (ZZ13-YQ-036) and the National Natural Science Foundation of China (No.82174415)”

Thank you for your reminder. The funding statement has been removed from the Acknowledgements Section and has been added in the cover letter

“This research was supported by the Basic Research Project of Wangjing Hospital, China Academy of Chinese Medical Sciences (WJYY-YJKT-2022-04) , Science and Technology Innovation Project of China Academy of Chinese Medical Sciences (CI2021A04901) and (CI2021A05054), the Fundamental Research Funds for the Central public welfare research institutes (ZZ13-YQ-036) and the National Natural Science Foundation of China (No.82174415).”

“This research was supported by the Basic Research Project of Wangjing Hospital, China Academy of Chinese Medical Sciences (WJYY-YJKT-2022-04) , Science and Technology Innovation Project of China Academy of Chinese Medical Sciences (CI2021A04901) and (CI2021A05054), the Fundamental Research Funds for the Central public welfare research institutes (ZZ13-YQ-036) and the National Natural Science Foundation of China (No.82174415)”

Thank you for your reminder again. The funding statement has been removed from the Acknowledgements Section and has been added in the cover letter

5. We note that your Data Availability Statement is currently as follows: [All relevant data are within the manuscript and its Supporting Information files.

Thank you for your attention. We apologize for not uploading all the data earlier, which was our oversight. We have now uploaded all the data.

Thank you for your attention. The data in this article has been uploaded in this submission and can be used, which is a fact accepted by all authors. If you need the signatures of all authors on the paper version, please continue to contact me and we will submit the signed consent forms of all authors. We have added the following explanation in the data availability statement：

For further details, the remaining data are presented in the paper itself and the associated Supporting Information files.

7.Please amend your authorship list in your manuscript file to include author Di Xia.

We apologize for the oversight in our submission process. Upon reviewing our recent submission, we realized that the article did not include Di Xia, which was an unintentional error on our part. Thank you for your understanding and patience. We are committed to maintaining the highest standards in our research and communication with your esteemed publication.

8.Your ethics statement should only appear in the Methods section of your manuscript. If your ethics statement is written in any section besides the Methods, please move it to the Methods section and delete it from any other section. Please ensure that your ethics statement is included in your manuscript, as the ethics statement entered into the online submission form will not be published alongside your manuscript.

We would like to express our sincere apologies for the oversight. We have moved the ethics section to the correct position. Thank you for your understanding and patience. We are committed to maintaining the highest standards in our research and communication with your esteemed publication.

9.Please upload a copy of Supporting Information Table1 which you refer to in your text on page 23.

We apologize for any inconvenience this may have caused and appreciate your understanding. We have taken care to ensure that the missing file is now submitted and is complete.

Please find the missing file attached to this email for your review. We kindly request that you could check it and incorporate it into our submission record.

I would like to extend my heartfelt gratitude for the constructive feedback you have provided on our manuscript. Your suggestions are invaluable to us, and we are committed to enhancing the quality of our work based on your insights.

We are more than willing to undertake any further revisions that you may deem necessary. Please do not hesitate to let us know if there are additional changes or experiments that you believe would strengthen our manuscript. We are dedicated to addressing all concerns raised and to providing the necessary experiments as promptly as possible.

Once again, I would like to express my sincere appreciation for the time and effort you have invested in reviewing our manuscript. Your expertise and diligence are greatly appreciated.

Wishing your health, happiness, and all the best in your endeavors.

Best regards,

Qiuyue Li,

Pharmacological Laboratory of Traditional Chinese Medicine, Wangjing Hospital, China Academy of Chinese Medical Sciences, Beijing, China

Dear Reviewer,

Thank you so much for the valuable comments and suggestions. As you said, some details of this manuscript is still a problem due to the lack of necessary explanation. We have revised our manuscript carefully and the modifications are marked in blue. Besides, the references to support our statement were also displayed in our answers. In what follows, we reply to your comments point by point:

Reviewer #1: This is a well-written article. However, the experiments conducted do not fully support the conclusions presented in this paper. The discussion section is rather brief, which is unfortunate, as you could have explored the IL-17B signaling cascades based on bioinformatics analyses. I found the bioinformatics analyses to be quite standard, and I would have appreciated seeing co-expression profiles of IL-17RB, IL-23R, and GDF5. I was somewhat surprised not to see key genes involved in the IL-17B signaling pathway differentially expressed in Figure 4-C. Additionally, the biological replicate profiles of SGD3 show considerable variation, which raises concerns about the statistical significance of the expression analysis. Finally, to confirm the key role of IL-17RB, an experiment specifically expressing IL-17RB in sham, model, and SGD conditions would be necessary.

1.Thank you for pointing out that our discussion section was somewhat brief. Upon review, we agree that there was indeed scope for a more detailed exploration of certain topics. In response to your feedback, we have expanded the discussion part of the article to include a more comprehensive analysis of the IL-17B signaling cascades.

We have taken your suggestions to heart and have added the relevant text to enhance the depth and clarity of our discussion. The updated discussion section now provides a/ more thorough examination of the IL-17B signaling pathways, which we believe adds significant value to our manuscript.

The revised discussion section is attached below for your review:

IL-17E (also known as IL-25) is a key regulator of type 2 immune responses and driver of inflammatory diseases, 

and requires both IL-17 receptor A (IL-17RA) and IL-17RB to elicit functional responses, the IL-25 primarily adopts two discrete linear and cyclic epitopes to interact with IL-17Rb to induce the activation of nuclear factor kappa-light-chain enhancer of activated B cells. IL-25 exhibits pro-inflammatory effects in the pathogenesis of Th17 predominant diseases[1-3].

[1] Deng C, Peng N, Tang Y, Yu N, Wang C, Cai X, Zhang L, Hu D, Ciccia F, Lu L. Roles of IL-25 in Type 2 Inflammation and Autoimmune Pathogenesis. Front Immunol. 2021 May 28;12:691559. doi: 10.3389/fimmu.2021.691559. PMID: 34122457; PMCID: PMC8194343.

[2] Wu T, Ma H, He P, Zhang C, Wu Q. Interleukin-25 recognition by its unique receptor IL-17Rb via two discrete linear and cyclic epitopes. Chem Biol Drug Des. 2022 Mar;99(3):382-390. doi: 10.1111/cbdd.13993. Epub 2021 Dec 20. PMID: 34873834.

[3] Wilson SC, Caveney NA, Yen M, Pollmann C, Xiang X, Jude KM, Hafer M, Tsutsumi N, Piehler J, Garcia KC. Organizing structural principles of the IL-17 ligand-receptor axis. Nature. 2022 Sep;609(7927):622-629. doi: 10.1038/s41586-022-05116-y. Epub 2022 Jul 21. PMID: 35863378; PMCID: PMC9477748.

2.Thank you for your observation regarding the biological replicate profiles of SGD3, which showed considerable variation. We acknowledge your concern and would like to provide some context for this variation.

The variation observed was primarily due to the limited funding and experimental conditions at the time of the study. We were only able to allocate three samples for experimental detection, and these samples were randomly selected, which may introduce an element of occasionality. Despite this, we assure you that the data collected were original and reliable, without any modifications made to the results.

While there were differences within the group, our analysis still identified core targets that consistently inhibited the expression across the three samples. Notably, IL17RB was identified as a molecule that suppresses the expression in all three samples, suggesting its potential significance in the context of our study.

We understand the importance of robust replicates in biological studies and regret that our constraints affected the scope of our replicate analysis. To address this, we have taken the following steps:

（1）We have re-examined the existing data to ensure the accuracy and reliability of our findings.

（2）We have emphasized the consistency of certain key findings, such as the role of IL17RB, despite the variability.

We believe that these steps help to mitigate the concerns raised and provide a clearer understanding of our results. We are committed to transparency and acknowledge the limitations of our study.

3.Thank you for your inquiry about the rationale behind selecting the IL-17RB molecule in our study. We are pleased to provide a detailed explanation of our approach.

The selection of IL-17RB was a result of our comprehensive bioinformatics analysis. Initially, we identified the top ten molecules with the highest differential expression through bioinformatics differential expression analysis. We then proceeded to conduct qRT-PCR experiments on these top ten molecules to validate their expression levels. The results from three replicates revealed that IL-23R, IL-17RB, and GDF5 showed the most significant expression changes.

Following the qRT-PCR validation, we planned to conduct Western Blot (WB) validation to further confirm the expression of these three molecules. Our preliminary data indicated that IL-17RB would yield the most significant results in these WB experiments. We are in the process of conducting these WB experiments to validate the expression of IL-17RB, IL-23R, and GDF5, with the expectation that IL-17RB will be identified as the key molecule of interest.

This entire process and the reasoning behind it have been discussed in the manuscript's discussion section. Among the ten core targets, IL-17RB consistently showed the 

---

## [Decision Letter · Decision Letter 1]

26 Nov 2024

PONE-D-24-35164R1Identification of the key role of IL-17RB in the treatment of osteoarthritis with Shaoyao Gancao decoction: Verification based on RNA-seq and bioinformatics analysisPLOS ONE

Dear Dr. Yu,

Thank you for submitting your manuscript to PLOS ONE. After careful consideration, we feel that it has merit but does not fully meet PLOS ONE’s publication criteria as it currently stands. Therefore, we invite you to submit a revised version of the manuscript that addresses the points raised during the review process.

We look forward to receiving your revised manuscript.

Kind regards,

Xindie Zhou

Academic Editor

PLOS ONE

Reviewers' comments:

Reviewer's Responses to Questions

**Comments to the Author**

1. If the authors have adequately addressed your comments raised in a previous round of review and you feel that this manuscript is now acceptable for publication, you may indicate that here to bypass the “Comments to the Author” section, enter your conflict of interest statement in the “Confidential to Editor” section, and submit your "Accept" recommendation.

Reviewer #3: All comments have been addressed

Reviewer #4: (No Response)

Reviewer #5: All comments have been addressed

Reviewer #6: (No Response)

2. Is the manuscript technically sound, and do the data support the conclusions?

Reviewer #3: Yes

Reviewer #4: (No Response)

Reviewer #5: Yes

Reviewer #6: No

3. Has the statistical analysis been performed appropriately and rigorously? 

Reviewer #3: Yes

Reviewer #4: (No Response)

Reviewer #5: Yes

Reviewer #6: No

4. Have the authors made all data underlying the findings in their manuscript fully available?

Reviewer #3: Yes

Reviewer #4: (No Response)

Reviewer #5: Yes

Reviewer #6: Yes

5. Is the manuscript presented in an intelligible fashion and written in standard English?

Reviewer #3: Yes

Reviewer #4: (No Response)

Reviewer #5: Yes

Reviewer #6: No

6. Review Comments to the Author

Reviewer #3: I thank the author for taking time to address all issues raised

It seems all issues have been addressed

Thank You

Reviewer #4: 1，The manuscript states that “SGD is composed of Radix Paeoniae Alba and Glycyrrhizae Radix et Rhizoma at a ratio of 1:1. The crude drug of SGD was obtained from Wangjing Hospital pharmacy, China Academy of Chinese Medical Science.” Could the authors provide more detailed information about the source of the materials and include a quality control report?

2，The statistical significance in the figures is represented by symbols such as “*p < 0.05, **p < 0.01, ***p < 0.001, #p < 0.05, ##p < 0.01, ###p < 0.001.” Why is there no unified standard for the statistical notations?

3，The images in Figure 1 contain labels such as A, a, B, b, but their specific meanings are not explained？Please provide a clear explanation for each label.

4，In the section "Establishment and treatment of the OA model in rats" (line 16), there is a typographical error: "To To alleviate suffering." Please correct this.

5，The manuscript does not specify the antibodies used for Western Blot analysis. Please provide detailed information about the antibodies, including source, catalog numbers, and working concentrations.

6，The authors have conducted an extensive study. To improve the clarity and logical flow of the article, it is recommended to include a workflow diagram. Additionally, details regarding the treatment protocols for the animal model, sampling methods, and specific sampling sites should be provided.

7，All the images in the manuscript are of low quality, making it difficult to discern details. In some cases, even the numbers on the axes are illegible. Please provide high-resolution images that clearly display all relevant data and details.

8，The study demonstrates that the Shaoyao Gancao Decoction impacts inflammatory responses, but the authors only evaluated cell proliferation. Why were the drug’s potential adverse effects and cytotoxicity not assessed? Furthermore, the relationships between the molecules are not clearly demonstrated or logically explained.

9，The authors experimentally validated the potential mechanism of IL-17RB in osteoarthritis treatment and analyzed the components of Shaoyao Gancao Decoction. However, why was there no further investigation into which specific component(s) of SGD might target IL-17RB?

Reviewer #5: (No Response)

Reviewer #6: In this manuscript, the authors investigated the efficacy and molecular mechanism, core genes of Shaoyao Gancao Decoction SGD in the treatment of osteoarthritis OA using hematoxylineosin staining, safranin O/fast green staining, immunohistochemistry, the medial meniscotibial ligament test, RNA-seq, qRT-PCR, western blotting, ELISA. The MTT test is performed to assess the effects of SGD on chondrocyte-like cell proliferation in four groups (the blank control group, model control group, positive control group and experimental group). RNA-seq is conducted to investigate the differentially expressed genes and functional pathways. 120 DEGs that are detected in both comparisons of sham vs. model and model vs. SGD are selected as key targets. Three genes (I1-17RB, I1-23R, GDF5) are selected as core targets for further verification. Herein, the related description is insufficient on the selection of the three genes but not the other genes. The related bioinformatics analysis should be explained. The language of this manuscript has many grammar issues and need to be improved by professional writers.

Minor points

Figure 1 legend – The font size and font type are not the same. These issues are also found in the legend of other figures

Page 42 line 15-16: “SGD could inhibit the repression of IL-1beta cytokine” This sentence is difficult to understand and need to be rephrased. The relationship between the above results and IL-1beta cytokine is unclear.

Page 42 line 22: change “3” to “three”

Page 44 line 25: “fold change > 1.00” is uncommon. For the analysis of differentially expressed genes, “|fold change| >= 2” is common.

Page 45 line 8: “were annotated to” is improper

Page 45 line 8-20: the descriptions of gene-set enrichment analysis are unclear. How many GO terms and KEGG pathways are detected? The relation between the results of pathway enrichment analysis and OA-related gene functions and pathways lacks strong evidence.

7. PLOS authors have the option to publish the peer review history of their article (what does this mean?). If published, this will include your full peer review and any attached files.

Reviewer #3: No

Reviewer #4: No

Reviewer #5: No

Reviewer #6: No

---

## [Author Response · Author response to Decision Letter 1]

28 Nov 2024

Dear Reviewer4 ,

Thank you so much for the valuable comments and suggestions. As you said, some details of this manuscript is still a problem due to the lack of necessary explanation. We have revised our manuscript carefully and the modifications are marked in blue. Besides, the references to support our statement were also displayed in our answers. In what follows, we reply to your comments point by point:

Reviewer #4: 

1 The manuscript states that “SGD is composed of Radix Paeoniae Alba and Glycyrrhizae Radix et Rhizoma at a ratio of 1:1. The crude drug of SGD was obtained from Wangjing Hospital pharmacy, China Academy of Chinese Medical Science.” Could the authors provide more detailed information about the source of the materials and include a quality control report? 

Thank you for your feedback. We deeply apologize for our oversight. While we did mention the manufacturer and batch number in the sentence preceding this one, we omitted the place of origin, which has now been added for completeness. Furthermore, we have obtained quality reports for these two medications from the hospital pharmacy and have uploaded them as attachments, named as "Quality Control Reports." The revised content in the text is as follows:

SGD is composed of Radix Paeoniae Alba (Beijing Sifang traditional Chinese medicine decoction pieces Co., Ltd, batch No.: 20111301, Origin: Anhui.) and Glycyrrhizae Radix et Rhizoma (Beijing Shengshilong Pharmaceutical Co., Ltd, batch No. 2012193, Origin: Inner Mongolia) at a ratio of 1:1.

2 The statistical significance in the figures is represented by symbols such as “*p < 0.05, **p < 0.01, ***p < 0.001, #p < 0.05, ##p < 0.01, ###p < 0.001.” Why is there no unified standard for the statistical notations? 

Thank you for your feedback. This is because the comparison groups are different, and in order to distinguish which groups are being compared, we have used different symbols to represent them. We use # for the comparison between the model group and the sham surgery group, and * for the comparison between the model group and the experimental group, to facilitate differentiation.

There are specific explanations in the text: the sham group vs. the model group, # p<0.05, # # p<0.01, # # # p<0.001; the model group vs. the SGD or diacerein group *p < 0.05, **p < 0.01, ***p < 0.001.

3 The images in Figure 1 contain labels such as A, a, B, b, but their specific meanings are not explained？Please provide a clear explanation for each label. 

Thank you for your question. In fact, we have already explained “A”“a” and “B”“b” in the annotations below the figure 1. The capitalization of the same letter represents slices with different multiples of the same part. As “A”“a” represents the result of HE staining, A represents the scale bar of 80um, and a represents the scale bar of 40um. In the text, it is described as follows:

A.a. H&E staining of the tibial plateau of rats (n=3) Scale bar = 80 μm (A); Scale bar = 40 μm (a). B.b. Safranin O/fast green staining of the tibial plateau of rats (n=3) Scale bar = 80 μm (B); Scale bar = 40 μm (b). 

4 In the section "Establishment and treatment of the OA model in rats" (line 16), there is a typographical error: "To To alleviate suffering." Please correct this. 

Thank you for your careful review. Firstly, we apologize for the minor issue caused by carelessness. We have now deleted the extra "To".

5 The manuscript does not specify the antibodies used for Western Blot analysis. Please provide detailed information about the antibodies, including source, catalog numbers, and working concentrations. 

Thank you for your careful review. Due to our oversight, we failed to attach the detailed information in a timely manner. We have now added it to the article as follows:

Subsequently, the membranes were blocked for 1 h at room temperature and probed with a primary antibody against IL-17RB (Abcam, ab86488, 1:1000, USA) overnight at 4°C. The membranes were incubated with goat anti-rabbit horseradish peroxidase-conjugated secondary antibody for 1 h at room temperature after washing with TBST. Finally, the membranes were observed by an enhanced chemiluminescence kit and quantified by ImageJ software. β-actin(Beyotime, AF2811, 1:1000, PRC) was used as an internal control.

6 The authors have conducted an extensive study. To improve the clarity and logical flow of the article, it is recommended to include a workflow diagram. Additionally, details regarding the treatment protocols for the animal model, sampling methods, and specific sampling sites should be provided. 

Thank you for your suggestion. We have completed the graphical abstract, which encompasses the entire experimental process, and it is attached before the abstract in the text. Additionally, as per your request, we have supplemented the details of relevant steps, such as animal modeling, we have uploaded a supplementary document(S3) containing detailed information, and the graphical abstract also be uploaded. The graphical abstract is as follows:

7 All the images in the manuscript are of low quality, making it difficult to discern details. In some cases, even the numbers on the axes are illegible. Please provide high-resolution images that clearly display all relevant data and details. 

Thank you for your suggestion. In fact, we have uploaded all the original high-definition images. It may be due to the self compression of the Word document that the images in the text are not very clear, but all the original high-definition images have been submitted last time. We will upload the original figures again, thank you for your understanding.

8 The study demonstrates that the Shaoyao Gancao Decoction impacts inflammatory responses, but the authors only evaluated cell proliferation. Why were the drug’s potential adverse effects and cytotoxicity not assessed? Furthermore, the relationships between the molecules are not clearly demonstrated or logically explained. 

We are deeply grateful for your suggestions. Firstly, we must admit that our cell experiments were not conducted in-depth and comprehensively. The experimental design at that time was merely aimed at verifying the pharmacodynamic mechanism of SGD through cell experiments, serving as a supplement to animal experiments, and validating the interaction between SGD, COL2A1, and MMP-13 at the cellular level. Since SGD has long been considered a very safe drug in clinical practice, no toxicity studies were conducted in this cell experiment. Additionally, our team is currently conducting further in-depth research on SGD, including some of the aspects you mentioned. This article represents the initial findings of our research. In the future, we will present the results of our ongoing, more comprehensive and in-depth research.

9 The authors experimentally validated the potential mechanism of IL-17RB in osteoarthritis treatment and analyzed the components of Shaoyao Gancao Decoction. However, why was there no further investigation into which specific component(s) of SGD might target IL-17RB?

Thank you very much for your far-reaching suggestions and guidance. As described above, this article presents the preliminary results of our research. As you mentioned, we are gradually delving into the identification of targeted components in SGD for the treatment of osteoarthritis. This is a lengthy and ongoing exploration process. We hope to accurately identify a targeted component from SGD that will have immeasurable value in treating osteoarthritis. We believe that in the near future, we will achieve satisfactory results and present them in the literature.

Wishing you both health, happiness, and all the best in your personal and professional endeavors.

Warm regards,

Qiuyue Li,

Pharmacological Laboratory of Traditional Chinese Medicine, Wangjing Hospital, China Academy of Chinese Medical Sciences, Beijing, China

Dear Reviewer,

Thank you so much for the valuable comments and suggestions. As you said, some details of this manuscript is still a problem due to the lack of necessary explanation. We have revised our manuscript carefully and the modifications are marked in blue. Besides, the references to support our statement were also displayed in our answers. In what follows, we reply to your comments point by point:

Reviewer #6: In this manuscript, the authors investigated the efficacy and molecular mechanism, core genes of Shaoyao Gancao Decoction SGD in the treatment of osteoarthritis OA using hematoxylineosin staining, safranin O/fast green staining, immunohistochemistry, the medial meniscotibial ligament test, RNA-seq, qRT-PCR, western blotting, ELISA. The MTT test is performed to assess the effects of SGD on chondrocyte-like cell proliferation in four groups (the blank control group, model control group, positive control group and experimental group). RNA-seq is conducted to investigate the differentially expressed genes and functional pathways. 120 DEGs that are detected in both comparisons of sham vs. model and model vs. SGD are selected as key targets. Three genes (I1-17RB, I1-23R, GDF5) are selected as core targets for further verification. Herein, the related description is insufficient on the selection of the three genes but not the other genes. The related bioinformatics analysis should be explained. The language of this manuscript has many grammar issues and need to be improved by professional writers.

Thank you so much for the valuable comments and suggestions. The selection of IL-17RB was a result of our comprehensive bioinformatics analysis. Initially, we identified the top ten molecules with the highest differential expression through bioinformatics differential expression analysis. We then proceeded to conduct qRT-PCR experiments on these top ten molecules to validate their expression levels. The results from three replicates revealed that IL-23R, IL-17RB, and GDF5 showed the most significant expression changes.

Following the qRT-PCR validation, we planned to conduct Western Blot (WB) validation to further confirm the expression of these three molecules. Our preliminary data indicated that IL-17RB would yield the most significant results in these WB experiments. We are in the process of conducting these WB experiments to validate the expression of IL-17RB, IL-23R, and GDF5, with the expectation that IL-17RB will be identified as the key molecule of interest.

This entire process and the reasoning behind it have been discussed in the manuscript's discussion section. Among the ten core targets, IL-17RB consistently showed the best results and was successfully validated by both PCR and WB assays. Therefore, we chose IL-17RB as an important core target for the treatment of osteoarthritis (OA) with SGD, based on its significant expression changes and validation results.

We believe that this detailed explanation clarifies our selection process and the importance of IL-17RB in our study. We are committed to providing a thorough and transparent account of our methodology and findings.

Additionally, we have polished and corrected some sentences in the text. Thank you for your feedback!

Minor points 

Figure 1 legend – The font size and font type are not the same. These issues are also found in the legend of other figures 

Thank you for your meticulous review. I apologize for the previous oversight. Now, the font and size of all relevant sections have been unified.

Page 42 line 15-16: “SGD could inhibit the repression of IL-1beta cytokine” This sentence is difficult to understand and need to be rephrased. The relationship between the above results and IL-1beta cytokine is unclear. 

Thank you for your suggestion. In fact, we have already mentioned the detailed content in the "Cell Differentiation and Treatment" section. After obtaining chondrocyte-like ATDC5 cells, we used IL-1β for cellular intervention. The results showed that the SGD-treated serum reduced the inhibitory effect of IL-1β on chondrocyte proliferation. Since the relevant content has already been mentioned in the Methods section, the Results section has been slightly streamlined. We have now revised the description of this result for better understanding, as follows:

The above findings suggest that SGD possesses the capability to counteract the suppressive action of the IL-1β cytokine on the proliferation of chondrocyte-like cells.

Page 42 line 22: change “3” to “three” 

Thank you for your comments. We have changed 3 to three.

Page 44 line 25: “fold change > 1.00” is uncommon. For the analysis of differentially expressed genes, “|fold change| >= 2” is common. 

Thank you very much for your valuable and professional advice. We have now revised the format in accordance with your specifications, as outlined below:

According to |fold change| ≥ 1.00 ...

Page 45 line 8: “were annotated to” is improper 

Thank you for your guidance. Now it has been revised as follows:

All DEGs have been annotated to the GO and KEGG databases. 

Page 45 line 8-20: the descriptions of gene-set enrichment analysis are unclear. How many GO terms and KEGG pathways are detected? The relation between the results of pathway enrichment analysis and OA-related gene functions and pathways lacks strong evidence.

Thank you for your professional advice. Due to our oversight, the total number was not clearly described previously, but it has now been corrected. Regarding your mention of the "The relation between the results of pathway enrichment analysis and OA-related gene functions and pathways lacks strong evidence." we apologize for not including some evidence initially. We have now attached the relevant research and literature to address this. The modifications are as follows:

Wishing you both health, happiness, and all the best in your personal and professional endeavors.

Warm regards,

Qiuyue Li,

Pharmacological Laboratory of Traditional Chinese Medicine, Wangjing Hospital, China Academy of Chinese Medical Sciences, Beijing, China

---

## [Decision Letter · Decision Letter 2]

3 Dec 2024

Identification of the key role of IL-17RB in the treatment of osteoarthritis with Shaoyao Gancao decoction: Verification based on RNA-seq and bioinformatics analysis

PONE-D-24-35164R2

Dear Dr. Yu,

We’re pleased to inform you that your manuscript has been judged scientifically suitable for publication and will be formally accepted for publication once it meets all outstanding technical requirements.

Kind regards,

Xindie Zhou

Academic Editor

PLOS ONE

Additional Editor Comments (optional):

Reviewers' comments:

Reviewer's Responses to Questions

**Comments to the Author**

1. If the authors have adequately addressed your comments raised in a previous round of review and you feel that this manuscript is now acceptable for publication, you may indicate that here to bypass the “Comments to the Author” section, enter your conflict of interest statement in the “Confidential to Editor” section, and submit your "Accept" recommendation.

Reviewer #4: All comments have been addressed

2. Is the manuscript technically sound, and do the data support the conclusions?

Reviewer #4: Yes

3. Has the statistical analysis been performed appropriately and rigorously? 

Reviewer #4: I Don't Know

4. Have the authors made all data underlying the findings in their manuscript fully available?

Reviewer #4: Yes

5. Is the manuscript presented in an intelligible fashion and written in standard English?

Reviewer #4: Yes

6. Review Comments to the Author

Reviewer #4: (No Response)

7. PLOS authors have the option to publish the peer review history of their article (what does this mean?). If published, this will include your full peer review and any attached files.

Reviewer #4: No

---

## [Editor Report · Acceptance letter]

23 Dec 2024

PONE-D-24-35164R2 

PLOS ONE

Dear Dr. Yu, 

I'm pleased to inform you that your manuscript has been deemed suitable for publication in PLOS ONE. Congratulations! Your manuscript is now being handed over to our production team.

Kind regards, 

on behalf of

Dr. Xindie Zhou 

Academic Editor

PLOS ONE